# Refining Hybrid Genetic Search for CVRP via Reinforcement Learning-Finetuned LLM

**Rongjie Zhu**[1,*]**, Cong Zhang**[2,*]**, Zhiguang Cao**[3]

[1]School of Teacher Education, Nanjing University of Information Science and Technology, China
[2]College of Computing and Data Science, Nanyang Technological University, Singapore
[3]School of Computing and Information Systems, Singapore Management University, Singapore
`rzhu114514@gmail.com`, `cong.zhang92@gmail.com`, `zgcao@smu.edu.sg`

## Abstract

While large language models (LLMs) are emerging as automated heuristic designers for solving vehicle routing problems (VRPs), state-of-the-art approaches predominantly rely on massive, general-purpose models like GPT-4. This work challenges this paradigm by demonstrating that smaller, specialized LLMs, when finely tuned, can generate components that surpass expert-designed heuristics within advanced solvers. We introduce RFTHGS, a novel **R**einforcement learning (RL) framework for **F**ine-**T**uning a small LLM to produce high-performance crossover (and subpopulation) operator for the **H**ybrid **G**enetic **S**earch (HGS) solver to solve the capacitated vehicle routing problem (CVRP). Our method utilizes a multi-tiered, curriculum-based reward function that progressively guides the LLM to first produce compilable code, then executable operators, and finally, components that exceed human expert-designed ones. Additionally, we introduce an operator caching mechanism to work in conjunction with the reward function, discouraging plagiarism and promoting diversity during training. Experimental results demonstrate that our fine-tuned LLM generates crossover (and subpopulation) operator which significantly outperforms those designed by human experts in HGS. This performance advantage is consistent, holding from small-scale instances and generalizing to large-scale problems of up to 1000 nodes. Furthermore, RFTHGS surpasses leading neuro-combinatorial baselines, prompt-based methods, and commercial LLMs, including GPT-4o and GPT-o4-mini.

## 1 Introduction

Combinatorial Optimization Problems (COPs) represent a fundamental class of computational challenges that arise across diverse domains including supply chain management, logistics, scheduling, and network design (Bengio et al., 2021). These problems, characterized by their discrete decision variables and complex constraints, are often NP-hard in complexity, making exact solutions computationally intractable for large-scale instances. For decades, researchers have developed specialized algorithms, heuristics, and metaheuristics to approximate optimal solutions, yet these approaches typically require significant domain expertise and manual design efforts (Papadimitriou & Steiglitz, 1998). The emergence of large language models (LLMs), with their remarkable reasoning and pattern recognition capabilities, has introduced a transformative paradigm for tackling COPs. By leveraging their natural language processing and generative abilities, LLMs enable automated approaches that reduce reliance on manual algorithm design and expert intervention (Sun et al., 2024; Novikov et al., 2025). The investigation of LLM-based approaches for solving Vehicle Routing Problems (VRPs) represents one of the most cutting-edge frontiers in this field.

Initial studies explored the use of LLMs as end-to-end solvers for VRPs (Yang et al., 2024). However, these purely generative approaches often yield solutions that are substantially inferior to those from conventional or deep learning-based solvers and are frequently infeasible, i.e., a shortcoming attributed to the fact that LLMs are notoriously prone to hallucinations (Kalai et al., 2025). Consequently, a more promising direction is to integrate LLMs not as autonomous solvers but as

---

*Both authors contributed equally

intelligent operators within established optimization frameworks, such as evolutionary algorithms. In this hybrid paradigm, the LLM acts as a strategic generator or refiner within an iterative loop. For instance, Liu et al. (2024c) employ general-purpose LLMs to guide an evolutionary process, using in-context prompting to perform crossover and mutation. An alternative approach inverts this relationship, using evolutionary computation not as a framework for the LLM to power, but as a mechanism to guide the LLM itself in generating increasingly effective heuristics. Representative works like EoH (Liu et al., 2024b) and ReEvo (Ye et al., 2024) iteratively refine LLM-generated heuristic through evolutionary selection. In contrast, another line of research prioritizes general-purpose frameworks for diverse VRP variants, favoring broader generalization at the cost of performance. Methods such as ARS (Li et al., 2025a), which leverage predefined structures to generate constraint-checking functions, and DRoC (Jiang et al., 2025), which utilizes retrieval to produce code that invokes external solvers like OR-Tools (Furnon & Perron, 2025), demonstrate improved generalization and robustness against code execution failures. Nonetheless, a significant performance gap remains with conventional and deep learning-based solvers, echoing doubts about the immediate application of existing methods to large-scale problems. Given that real-world instances rely on advanced solvers, a critical question is whether we can finetune small LLMs to optimize key components within these solvers to achieve beyond expert performance, presenting a challenging yet promising research frontier.

We introduce RFTHGS, a reinforcement learning (RL) framework for fine-tuning a reasoning LLM with 14B parameters to autonomously generate effective crossover (and subpopulation) operator for the Hybrid Genetic Search (HGS) algorithm (Vidal, 2022), thereby enhancing its performance in solving large-scale Capacitated Vehicle Routing Problems (CVRP). Our RL approach utilizes solution quality as the key feedback signal, instantiated as a structured, tiered reward design. This reward is designed to guide the learning process through three progressive stages. First, since instruction-tuned models often fail to produce syntactically valid code, we reward compilable outputs. Second, an additional reward is granted if the operator code executes successfully without runtime errors or timeouts. Finally, for the compilable and executable code, the relative improvement in solution quality on a predefined set of CVRP instances compared to a baseline expert-designed operator, translates into a linear reward (positive for outperforming the baseline, negative otherwise). It is important to emphasize that the CVRP instances serve exclusively to steer the feedback mechanism by evaluating operators generated by the LLM and are not provided as input to the model itself. To prevent reward hacking and repeated generation of the same high-performing operator, we incorporate an operator buffer mechanism. This mechanism penalizes the model for producing duplicates, thereby explicitly incentivizing diversity in the discovered solutions. Through this iterative refinement process, our framework empowers the LLM-generated operators to ultimately exceed the performance of handcrafted operators designed by human experts. Extensive experiments verify that the LLM-generated crossover (and subpopulation) operator delivers substantial improvements over the expert-designed operator in HGS, achieving superior performance on both small and large-scale instances (up to 1,000 nodes) on real-world benchmark. Furthermore, it surpasses all leading neuro-combinatorial and prompt-based LLM baselines by a significant margin. This work provides, to the best of our knowledge, the first evidence that a small-scale reasoning LLM (14B parameters) can be fine-tuned via RL to produce critical components that exceed the performance of those in state-of-the-art, expert-designed solvers.

## 2 RELATED WORK

### 2.1 ON THE REASONING ABILITY OF LARGE LANGUAGE MODELS

The development of large language models (LLMs) with advanced reasoning capabilities has evolved through several key phases, beginning with Chain of Thought (CoT) prompting, which explicitly guides models to generate intermediate reasoning steps, significantly improving performance on tasks like arithmetic and commonsense reasoning (Plaat et al., 2024; Wei et al., 2022). This approach was further enhanced by inference-time strategies such as Self-Consistency (aggregating multiple reasoning paths) and Tree-of-Thoughts (ToT) (exploring branched reasoning trajectories), which reduce errors and improve robustness in multi-step problem-solving (Yao et al., 2023). A major shift occurred with the integration of reinforcement learning (RL), where models are trained using verifiable rewards (e.g., correct answers in mathematical problems or code execution results) to incentivize logical reasoning without relying solely on supervised fine-tuning (Xiang

et al., 2025; Xu et al., 2025). For instance, DeepSeek-R1 (DeepSeek-AI et al., 2025) and OpenAI's o1 series (OpenAI et al., 2024) exemplify how RL-driven self-improvement and scaled inference-time compute enable deliberate, step-by-step reasoning. Additionally, hybrid methods such as Microsoft's rStar-Math (Guan et al., 2025) (which integrates Monte Carlo Tree Search for problem decomposition) and retrieval-augmented generation (RAG) (Lewis et al., 2020) combine the pattern recognition capabilities of LLMs with external tools for rigorous symbolic operations. Recent advancements also focus on test-time training and outcome-based exploration to enhance adaptability and diversity in reasoning paths, while interpretability research aims to ensure faithful internal reasoning processes (Song et al., 2025). Despite progress, challenges such as hallucination, scalability, and generalisation persist, driving ongoing innovation in architectures and training paradigms (Shojaee et al., 2025). In contrast to the well-recognized success in solving math problems, training the LLM with reasoning capabilities to generate operators that can outperform the default expert-designed ones in advanced VRP solvers remains challenging and largely unexplored.

## 2.2 Solving CVRP With LLM In The Loop

The integration of LLMs into Vehicle Routing Problems (VRPs) has primarily advanced through prompting-based methodologies (Yang et al., 2024; Jiang et al., 2025; Liu et al., 2024c; Huang et al., 2024), which leverage the robust reasoning capabilities of state-of-the-art (SOTA) off-the-shelf LLMs (e.g., GPT-o3-mini). For instance, ARS (Li et al., 2025b) uses LLMs to automatically generate constraint-aware heuristics for solving complex vehicle routing problems by synthesizing natural language descriptions into executable code, which can construct heuristics for 90% of common VRP variants with different constraints. Similarly, Hercules (Wu et al., 2025) employs Core Abstraction Prompting (CAP) to derive high-performance heuristics by abstracting core components from elite solutions, though it remains dependent on powerful closed-source LLMs. Recent work has explored automatic heuristic design using LLMs. Representative works like EoH (Liu et al., 2024b) and ReEvo (Ye et al., 2024) iteratively refine LLM-generated heuristics through evolutionary selection. Most recently, CALM (Huang et al., 2025) extends this paradigm by integrating reinforcement fine-tuning of the LLM into the evolutionary loop, allowing the model and its generated heuristics to co-evolve. In contrast, finetuning-based approaches for COP remain relatively sparse, often due to challenges like catastrophic forgetting, computational costs, and overfitting when adapting pre-trained models to specialized domains. While parameter-efficient methods like LoRA (Hu et al., 2021) mitigate some issues, fine-tuning small open-source LLMs (e.g., LLaMA (Touvron et al., 2023)) to generate operators for widely-used solvers like Hybrid Genetic Search (HGS) (Vidal, 2022) remains an open problem. Current efforts focus primarily on prompting, leaving a gap in developing specialized, lightweight models that can efficiently integrate with solver frameworks without relying on API-dependent, proprietary LLMs.

## 3 Preliminary

**HGS For Solving The Capacitated Vehicle Routing Problem (CVRP).** The Hybrid Genetic Search (HGS) (Vidal, 2022) algorithm represents a state-of-the-art metaheuristic framework prominently applied to complex combinatorial optimization problems, particularly vehicle routing problems (VRPs). As an extension of the classical genetic algorithm, HGS distinguishes itself through a tight integration of population-based evolutionary search and intensive local improvement procedures. Its core mechanism involves maintaining a diverse population of solutions that are iteratively refined through a process of selection, crossover, and local search. Within this framework, the crossover operator is the primary mechanism for global exploration by recombining genetic material from parent solutions to generate novel offspring. Rather than producing trivial combinations, it constructs high-quality solution skeletons that effectively inherit desirable attributes from both parents. These offspring solutions subsequently undergo rigorous local search, which acts upon the foundation laid by crossover to exploit the solution space locally and achieve feasibility and optimality. This synergistic interplay, where crossover provides a robust starting point for deep local exploitation, is a critical factor in the documented efficacy of HGS, enabling it to navigate the trade-off between exploration and exploitation effectively and consistently produce high-quality solutions for routing problems.

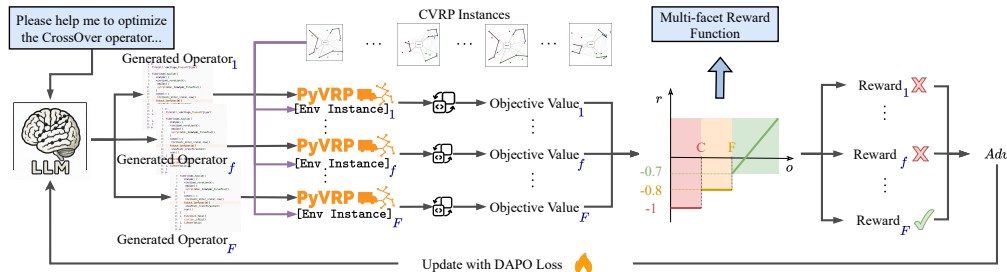

Figure 1: **The reinforcement learning pipeline of RFTHGS.** The framework iteratively optimizes an LLM to generate effective crossover operators for HGS. Each iteration consists of generating code from a structured prompt, evaluating the operator's performance on a validation set (using incremental compilation for speed), calculating a multi-faceted reward, and updating the LLM policy. The LLM only sees operator examples, not problem instances or the solver codebase.

**The Group Relative Policy Optimization (GRPO) Algorithm**. GRPO (Shao et al., 2024) is an improved variant of Proximal Policy Optimization (PPO) (Schulman et al., 2017). The key innovation of GRPO lies in utilizing a normalized reward function to compute advantages, where the mean and variance are estimated through Monte-Carlo sampling (with sample size $G$) from the current policy $\pi_k(\cdot|x)$ at step $k$ for each input (prompt) $x$. For given parameters $\epsilon$, $\beta > 0$, and a reference policy $\pi_{ref}$ (usually the base model), the GRPO objective optimization problem is formulated as:

$$\max_{\pi} \mathbb{E}_{y \sim \pi_k(\cdot|x)} \min \left[ \frac{\pi(y|x)}{\pi_k(y|x)} A_{\pi_k}(x, y), \text{clip}\left( \frac{\pi(y|x)}{\pi_k(y|x)}, 1 - \varepsilon, 1 + \varepsilon \right) A_{\pi_k}(x, y) \right] - \beta \text{KL}(\pi||\pi_{\text{ref}})$$

(1)

where KL denotes Kullback-Leibler divergence, and $A_{\pi_k}$ represents GRPO advantage function:

$$A_{\pi_k}(x, y_i) = \frac{r(x, y_i) - \mathbb{E}_{\pi_k} r(x, y_i)}{\sqrt{\mathbb{E}_{\pi_k}(r(x, y_i) - \mathbb{E}_{\pi_k} r(x, y_i))^2 + \varepsilon}} \simeq \frac{r(x, y_i) - \mu(\{r_\ell\})}{\sqrt{\sigma^2(\{r_\ell\}) + \varepsilon}}, \quad 1 \leq \ell \leq G$$

(2)

with the advantage estimated by sampling a "group" of size $G$ for each input $x$, and $\mu$ and $\sigma$ represent the empirical mean and standard deviation, respectively.

## 4  METHOD

We introduce RFTHGS, a reinforcement learning framework that fine-tunes large language models (LLMs) to generate crossover operators that outperform expert-designed ones in the Hybrid Genetic Search (HGS) solver (Vidal, 2022). The framework uses solution quality as the key reward signal to guide the LLM toward generating increasingly effective operators. As shown in Figure 1, RFTHGS is an iterative closed feedback loop. Each iteration consists of prompting the LLM to generate new operators, assessing their performance on a set of predefined CVRP instances, and then employing the feedback reward to refine the LLM via reinforcement learning.

Specifically, each iteration begins by constructing a few-shot CoT context (see Appendix A.3) that contains: (1) instructions specifying key properties (e.g., diversity and quality) and steps for generating a high-quality operator, and (2) examples of existing operators that illustrate the required structure and syntax. The LLM then generates new crossover operators based exclusively on this prompt, with no references to other operators (or modules) in the HGS library or access to specific CVRP instances. For evaluation, each generated operator is integrated into the HGS library, and the code is recompiled to test on a fixed problem benchmark set. We employ incremental compilation to speed up this recompilation step. Finally, performance metrics such as compilability and improvement over baseline operators are combined into a multi-faceted reward, which is used to finetune the LLM via reinforcement learning. The RFTHGS framework automates the design of optimization operators, and demonstrates that small LLMs can evolve components that outperform

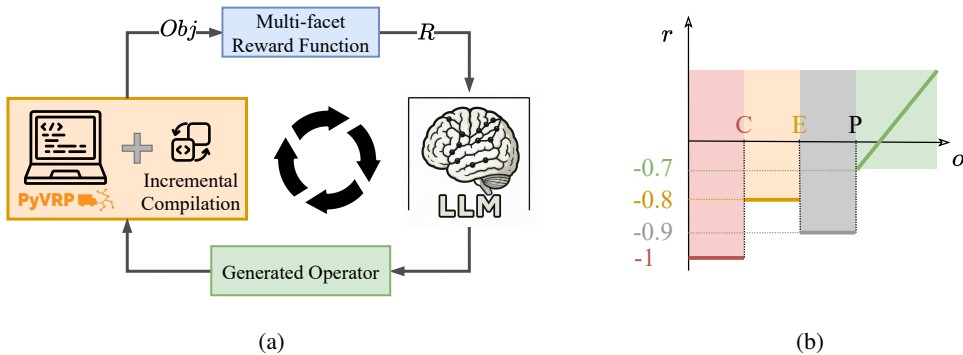

(a)            (b)

Figure 2: (a) HGS as the environment for evaluating the quality of LLM-generated operators. We use the incremental compilation technique to boost the computation of objective values. (b) The multi-faceted reward function.

human-designed ones in a state-of-the-art CVRP solver. We also verify that the RFTHGS framework is generic and can be applied to optimize other operators (or modules) within HGS, as demonstrated in Appendix A.5, where we also apply RFTHGS to optimize the subpopulation operator as another example.

## 4.1 ONE-STEP POMDP MODELLING

We formulate the operator optimization as a one-step Partially Observable Markov Decision Process (One-Step POMDP), with formal definitions of state, action, reward and policy given as follows.

**State**: The state $X \in \mathcal{X}$[1] denotes a tokenized prompt received by the LLM, comprising both the task instructions and examples of the target operator to be optimized. To mitigate context overload, the input is restricted to the target operator itself (e.g., the crossover operator in HGS), rather than the entire library of the solver. This restriction results in a partially observable environment, as the model only perceives a subset of the full state space (i.e., the complete solver repository).

To enhance the versatility of the learning process, we maintain a buffer of few-shot examples containing operators generated by the LLM during training as well as those designed by human experts. At each iteration, we randomly sample examples from this buffer to construct the prompt. This approach enables the LLM to learn from and attempt to improve upon both its previous generations and expert-designed operators. Furthermore, it enriches the diversity of the initial states (i.e., prompts), which helps prevent overfitting and encourages broader exploration.

**Action**: The action (or response) $Y \sim p(\cdot | X \in \mathcal{X})$ generated by our operator-refining LLM is a sequence of tokens $Y \in \mathcal{Y}$ consisting of two main parts, where $p(\cdot | X)$ is the conditional probability from which the actions are sampled. The first part is a reasoning segment enclosed between the special tokens `<think>` and `</think>`, where it explains the steps and logic it plans to take for the optimization task. Following this, it outputs optimized version of the code for the target operator.

**State Transition**: The state terminates after the action is generated, since we only allow one round of optimization for the operator. Therefore, there are no state transitions, and the POMDP is one-step.

**Reward**: The reward $r \in \mathbb{R}$ is a scalar evaluating the quality of LLM-generated operators. Please refer to Section 4.2 below for details of the reward function.

**Policy Network**: The policy network is a base large language model, denoted by $\pi_\theta(\cdot | X)$, $X \in \mathcal{X}$, with trainable parameters $\theta$ that parameterize the conditional probability distribution $p(\cdot | X)$ from which the optimized operator is sampled. In this work, we focus on relatively small LLMs (e.g., 14B parameters), which can be either a pretrained base model or an instruction-tuned variant.

---

[1]Here we use the symbol $X \in \mathcal{X}$ to represent the state to highlight that the RL task here is different from the conventional RL ones, where the initial state is the input to the LLM.

Table 1: **Performance comparison of baselines and our method for CVRPLIB across problem sizes.** Light gray columns indicate generalization to unseen problem sizes, while light gray rows represent generalization to higher iterations. The darker gray intersection areas highlight double generalization across both dimensions. Bold values denote best performance among all methods; asterisks (*) indicate that the results are unavailable.

| Methods | $n \in [100, 200)$ Gap% (↓) | Time (s) | $n \in [200, 400)$ Gap% (↓) | Time (s) | $n \in [400, 600)$ Gap% (↓) | Time (s) | $n \in [600, 800)$ Gap% (↓) | Time (s) | $n \in [800, 1000]$ Gap% (↓) | Time (s) |
|---|---|---|---|---|---|---|---|---|---|---|
| **Conventional Solver** | | | | | | | | | | |
| HGS-PyVRP$_{800}$ (Wouda et al., 2024) | 0.62 | 12.45 | 1.85 | 28.16 | 1.95 | 58.41 | 2.62 | 91.31 | 2.32 | 121.04 |
| HGS-PyVRP$_{1000}$ (Wouda et al., 2024) | 0.55 | 14.88 | 1.66 | 36.27 | 1.81 | 72.86 | 2.43 | 110.54 | 2.22 | 144.17 |
| HGS-PyVRP$_{1200}$ (Wouda et al., 2024) | 0.52 | 17.25 | 1.56 | 42.98 | **1.69** | 88.40 | 2.32 | 129.58 | 2.10 | 173.15 |
| OR-Tools (Furnon & Perron, 2025) | 4.26 | 88.83 | 5.05 | 172.07 | 4.98 | 296.53 | 6.71 | 416.95 | 4.65 | 532.32 |
| LKH (Helsgaun, 2000) | 1.42 | 191.12 | 1.97 | 252.23 | 2.85 | 432.36 | 3.65 | 599.41 | 3.31 | 545.85 |
| **NCO** | | | | | | | | | | |
| POMO (Kwon et al., 2020) | 13.30 | 0.41 | 14.64 | 0.64 | 22.07 | 1.29 | 21.57 | 2.32 | 41.23 | 4.16 |
| MTPOMO (Liu et al., 2024a) | 6.50 | 0.98 | 8.79 | 1.02 | 16.58 | 1.89 | 26.56 | 2.81 | 28.19 | 4.33 |
| MVMoE (Zhou et al., 2024) | 5.46 | 0.80 | 8.14 | 1.56 | 13.26 | 2.85 | 16.59 | 4.17 | 18.40 | 6.25 |
| RF-POMO (Berto et al., 2025) | 5.67 | 0.54 | 7.07 | 1.15 | 10.29 | 1.97 | 12.28 | 2.86 | 13.31 | 4.48 |
| RF-MoE-L (Berto et al., 2025) | 7.15 | 0.85 | 7.67 | 1.58 | 10.76 | 2.79 | 15.15 | 3.95 | 15.70 | 5.84 |
| AM (Kool et al., 2019) | 200.75 | 0.30 | 204.59 | 0.58 | 253.98 | 1.12 | 301.08 | 1.51 | 280.49 | 2.01 |
| DeepACO (Ye et al., 2023) | 76.18 | 18.37 | 93.02 | 35.03 | 97.70 | 61.03 | 123.89 | 88.23 | 116.13 | 112.01 |
| NeuroLKH (Xin et al., 2021) | 1.96 | 1.02 | * | * | * | * | * | * | * | * |
| NeuOpt (Ma et al., 2023) | 3.51 | * | * | * | * | * | * | * | * | * |
| **Prompting-Based Method With LLM** | | | | | | | | | | |
| MCTS-AHD (Zheng et al., 2025) | 18.51 | 4.35 | 19.07 | 10.06 | 18.40 | 22.50 | 28.51 | 37.60 | 19.70 | 55.61 |
| ReEvo (Ye et al., 2024) | 72.11 | 4.96 | 96.55 | 10.78 | 107.40 | 23.99 | 163.62 | 41.72 | 144.22 | 61.49 |
| GPT4o$_{800}$ (Hurst et al., 2024) | 0.62 | 12.1 | 1.85 | 29.3 | 1.95 | 59.4 | 2.62 | 91.7 | 2.32 | 119.2 |
| GPT4o$_{1000}$ (Hurst et al., 2024) | 0.55 | 15.0 | 1.66 | 36.1 | 1.81 | 73.4 | 2.43 | 111.2 | 2.22 | 143.7 |
| GPT4o$_{1200}$ (Hurst et al., 2024) | 0.52 | 17.7 | 1.56 | 43.1 | 1.69 | 87.3 | 2.32 | 130.5 | 2.10 | 172.8 |
| GPT-o3$_{800}$ (Jaech et al., 2024) | 0.62 | 11.96 | 1.85 | 30.20 | 1.95 | 58.94 | 2.62 | 91.39 | 2.32 | 119.04 |
| GPT-o3$_{1000}$ (Jaech et al., 2024) | 0.55 | 14.31 | 1.66 | 34.80 | 1.81 | 73.61 | 2.43 | 109.78 | 2.22 | 143.19 |
| GPT-o3$_{1200}$ (Jaech et al., 2024) | 0.52 | 17.65 | 1.56 | 43.53 | 1.69 | 87.89 | 2.32 | 130.71 | 2.10 | 172.93 |
| GPT-o4-mini$_{800}$ (Jaech et al., 2024) | 0.70 | 10.9 | 1.78 | 26.1 | 1.99 | 54.4 | 2.80 | 86.7 | 2.28 | 114.8 |
| GPT-o4-mini$_{1000}$ (Jaech et al., 2024) | 0.63 | 13.7 | 1.66 | 32.3 | 1.86 | 66.8 | 2.70 | 106.3 | 2.20 | 140.6 |
| GPT-o4-mini$_{1200}$ (Jaech et al., 2024) | 0.58 | 16.3 | 1.58 | 38.6 | 1.74 | 79.4 | 2.61 | 127.1 | 2.11 | 169.6 |
| **Ours** | | | | | | | | | | |
| RFTHGS$_{800}$ | 0.70 | 13.14 | 1.67 | 29.60 | 1.83 | 61.65 | 2.59 | 92.17 | 2.24 | 118.59 |
| RFTHGS$_{1000}$ | 0.52 | 14.33 | 1.62 | 36.16 | 1.76 | 74.16 | 2.35 | 110.43 | 2.17 | 143.87 |
| RFTHGS$_{1200}$ | **0.46** | 19.13 | **1.55** | 44.18 | 1.73 | 87.84 | **2.26** | 132.04 | **2.09** | 172.56 |

## 4.2 THE MULTI-FACETED REWARD DESIGN WITH ANTI-PLAGIARISM CACHE

Building on the insight that carefully crafted, multi-faceted rewards are crucial for effective RL (Narvekar et al., 2020; Eppe et al., 2022; Huang et al., 2025), we developed a multi-tiered reward function to decompose the learning process. Particularly, the reward function follows a curriculum learning principle, guiding the LLM through progressive stages to evolve operators that exceed those designed by human experts. To ensure the robustness of this approach, we further introduce two key innovations: a mechanism to prevent reward hacking by deterring plagiarism of prompt examples, and a method to significantly accelerate the training process.

**Anti-Plagiarism Cache With Abstract Syntax Tree**. To mitigate reward hacking and encourage the exploration of unseen operators, we introduce a caching mechanism that leverages Abstract Syntax Trees (ASTs) to deter plagiarism. The AST provides a structured, hierarchical representation that abstracts away unnecessary syntactic details like punctuation and formatting to capture the essential logical structure of the generated operators. We cache the AST representations of all few-shot operator examples in the prompt. For each operator generated by the policy $\pi_\theta$, its AST is compared against those in the cache. A penalty is invoked by the reward function if a substantial match is detected, indicating direct copying. This approach promotes diverse exploration by penalizing redundant operator generation. Appendix A.7 presents the details of the AST comparison mechanism.

**HGS With Incremental Compilation As The Evaluator**. We have to integrate each generated operator into the HGS library to evaluate its quality. This process inevitably requires recompiling the

repository, which will incur prohibitive computational overhead, especially for large training batch sizes. Nonetheless, recompiling the entire library is unwarranted when only a single, small code snippet (the generated operator) is modified. To address this bottleneck, we employ an incremental compilation technique that selectively recompiles only the modified code and its dependencies, reducing recompilation time to approximately 25% of compiling the whole library and significantly accelerating the training speed.

Here we give the formulation of our three-stage reward function. First, the reward function assigns a reward of $-0.8$ for a syntactically correct and compilable operator to encourage a rapid transition from invalid code and improve exploration efficiency, or a penalty of $-1$ for invalid output. Upon achieving compilability, the function then assesses executability, penalizing runtime failures such as timeouts. An operator that executes successfully receives a reward of $-0.7$, independent of its solution quality. Finally, for executable operators, performance is evaluated on a predefined set of CVRP instances, with the reward quantified as the relative improvement over expert-designed benchmarks according to the following calculation:

$$
r(o) = \begin{cases} -1 & o \notin \mathrm{C} \\ -0.8 & o \in \mathrm{C}, o \notin \mathrm{E} \\ -0.9 & o \in \mathrm{C}, o \in \mathrm{E}, o \in \mathrm{P} \\ \max\left(-0.7, [\phi_{\mathrm{HGS}}^{J}(o_{\mathrm{expert}}) - \phi_{\mathrm{HGS}}^{J}(o)] / \phi_{\mathrm{HGS}}^{J}(o_{\mathrm{expert}})\right) & o \in \mathrm{C}, o \in \mathrm{E}, o \notin \mathrm{P} \end{cases} \tag{3}
$$

In this formulation, $o$ represents the generated operator, while C, E, and P correspond to the sets of compilable, executable, and plagiarized code, respectively. For evaluation, the HGS library is recompiled to include the generated operator $o$. The performance metric $\phi_{\mathrm{HGS}}^{J}(\cdot)$ is then calculated as the average result on $J$ random CVRP instances. To benchmark the effectiveness of our continuous reward design in Equation 3, we compare it with a discrete version where we employ a +1 reward if the generated operator outperforms the baseline operator in Table 3 (details in Appendix 5.4). We find that our continuous reward offers feedback proportional to performance gains, enabling sustained refinement and explaining its superior performance.

## 4.3 THE REINFORCEMENT LEARNING ALGORITHM

We use DAPO (Yu et al., 2025) as the reinforcement learning algorithm for training our operator refining network. Specifically, DAPO is an improved version of GRPO with four adjustments: 1). *Clip-Higher Mechanism*. Unlike GRPO following the original PPO setting where a unified clip ration is adopted for the positive and negative responses, DAPO decouples the clipping range into a higher upper bound ($\varepsilon_{\mathrm{high}}$) and a standard lower bound ($\varepsilon_{\mathrm{low}}$), allowing the policy to more aggressively increase probabilities for promising but initially low-likelihood tokens. This promotes greater exploration and diversity in generated responses, effectively preventing entropy collapse where the model becomes overly deterministic; 2). *Dynamic Sampling*. This strategy filters out prompt groups where all sampled responses are either all correct or all incorrect, as these yield zero advantage and provide no learning signal. By replacing them with new prompts that exhibit varied performance, DAPO ensures every training batch contains meaningful gradients, improving training efficiency and stability without sacrificing throughput. However, in our paper, we deprecate this design as the reward signal in our case is continuous, specifying a wide range of situations from uncompilable code to superior performance gain against the baseline operators that all contribute useful learning signals for the LLM to learn; 3). *Token-Level Policy Gradient Loss*. Unlike GRPO, which averages losses at the response level, DAPO calculates and aggregates the loss over all tokens in the batch before averaging. This ensures each token's contribution to the gradient is weighted equally, providing more precise updates for long reasoning chains and better reinforcing correct steps in lengthy responses; 4). *Overlong Reward Shaping*. To address the issue of truncated lengthy responses that may contain valid reasoning, DAPO employs two strategies: Overlong Filtering excludes these responses from training updates to avoid misleading penalties, and Soft Overlong Punishment applies a gradual, length-dependent penalty beyond a certain token threshold to encourage conciseness without harshly punishing correct but verbose reasoning. The pseudo-code of our algorithm is shown in Algorithm 1.

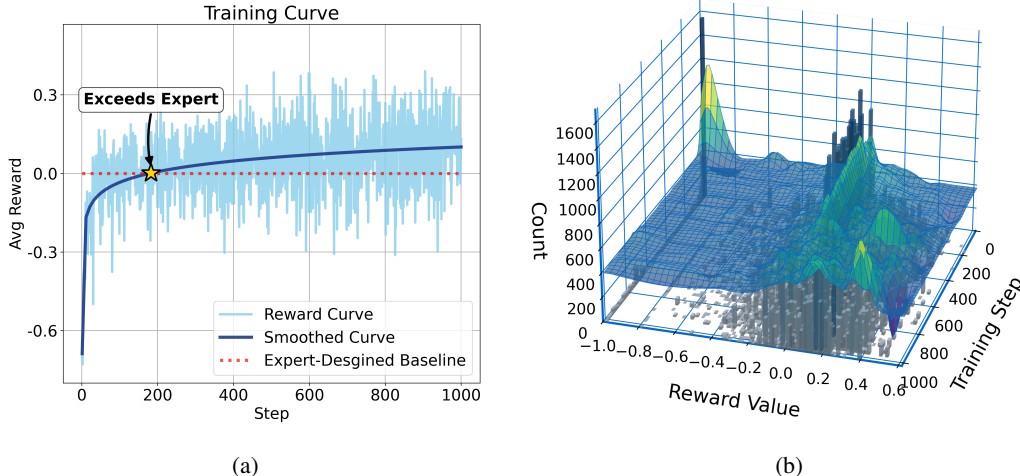

Figure 3: **Training dynamics of the RFTHGS framework.** (a) Average reward per step, showing stable convergence. (b) Evolution of the reward distribution, illustrating the effectiveness of the multi-faceted reward function in guiding the learning process.

## 5 EXPERIMENTS

### 5.1 EXPERIMENT SETTINGS AND BASELINES

We give the details of the configurations of our RFTHGS algorithm. Specifically, we initialize the policy with Qwen-14B reasoning LLM (Yang et al., 2025). For DAPO, we follow its optimal settings reported in the original paper with $\varepsilon_{\text{high}} = 0.28$ and $\varepsilon_{\text{low}} = 0.2$. The batch size is set to 16 and the rollout group size is also set to 16. Therefore, the policy model will generate 256 crossover operators for each step. For calculating reward during training, we use a fixed set of 30 CVRP instances sampled from the CVRPLIB X instances (Uchoa et al., 2017), restricting the selection to those with at most 400 nodes. During the testing phase, we sample 16 operators and report the performance of the best one. The final evaluation of our method is performed on the CVRPLIB benchmark, which encompasses a wide range of instance sizes from small scales to industry-level scales (up to 1000 nodes). We benchmark RFTHGS against a variety of baselines on CVRPLIB X instances (Uchoa et al., 2017). These include the state-of-the-art conventional solvers, neuro-combinatorial techniques, and prompting strategies that utilize commercial LLMs such as the GPT-4 series. To ensure an equitable comparison for the LLM-based approaches, we consistently sample 16 operators and select the best one for each. Further details on the baselines are available in Table 1. Detailed breakdown of the computational resources required for RL training is provided in Appendix A.6

### 5.2 PERFORMANCE ON CVRPLIB

Table 1 compares RFTHGS against a diverse set of baselines on CVRPLIB instances, including conventional heuristics, neuro-combinatorial methods, and prompting techniques that utilize commercial LLMs such as the flagship GPT-4o series. The results demonstrate that RFTHGS outperforms all

Table 2: **Successful compilation rate**.

| Successful Compilation Rate | | | |
|---|---|---|---|
| GPT-4o | GPT-o3 | GPT-o4-mini | RFTHGS-14B |
| 3/16 | 9/16 | 3/16 | **16/16** |

baseline methods by a substantial margin. This superior performance is underscored by its exceptional generalization capability to large-scale problems unseen during training. Notably, although trained exclusively on instances with $n < 400$, our approach generalizes effectively to instances of up to $n = 1000$, which are more than twice the size of the largest training instances. This validates the potential of refining advanced solvers via learned components for complex combinatorial optimization problems. The code comparison between human-expert designed operators and

Table 3: **Ablation study on reward design.** FRTHGS$_d$ is the 14B LLM trained with reward in Equation 4. FRTHGS$_c$ is the 14B LLM trained with reward in Equation 3. Shaded areas are generalization results.

| Methods | $n \in [100, 200)$ | | $n \in [200, 400)$ | | $n \in [400, 600)$ | | $n \in [600, 800)$ | | $n \in [800, 1000]$ | |
| --- | --- | --- | --- | --- | --- | --- | --- | --- | --- | --- |
| | Gap% ($\downarrow$) | Time (s) | Gap% ($\downarrow$) | Time (s) | Gap% ($\downarrow$) | Time (s) | Gap% ($\downarrow$) | Time (s) | Gap% ($\downarrow$) | Time (s) |
| HGS | **0.62** | 12.45 | 1.85 | 28.16 | 1.95 | 58.41 | 2.62 | 91.31 | 2.32 | 121.04 |
| FRTHGS$_d$ | 0.83 | 11.75 | 1.78 | 28.89 | 1.92 | 58.12 | 2.64 | 94.30 | 2.30 | 120.28 |
| FRTHGS$_c$ | 0.70 | 13.14 | **1.67** | 29.60 | **1.83** | 61.65 | **2.59** | 92.17 | **2.24** | 118.59 |

our LLM-optimized operators is presented in Supplementary Material, highlighting the key modifications introduced by the LLM that contribute to improved performance. We also compare with COMPASS (Chalumeau et al., 2023), a novel state-of-the-art NCO method for solving VRPs. The results in Appendix A.8 shows the consistent superior performance of RFTHGS against COMPASS.

Another key observation is that our RFTHGS framework enables a 14B-parameter LLM to outperform trillion-parameter GPT reasoning models (GPT-4o, GPT-o3, GPT-o4-mini). This advantage is demonstrated through both the quality of the modifications and their practical efficacy. As shown in Table 2, our model achieves a perfect successful compilation rate of 16/16, substantially exceeding the rates of the GPT models (3/16, 9/16, and 3/16, respectively). Crucially, while the GPT models often introduce numerous modifications, these changes consistently fail to improve performance. This is evident in Table 1, where the crossover operators modified by these GPT models exhibit performance identical to the original, unmodified operator, confirming that no functionally helpful modifications were made. In contrast, our RFTHGS-guided model produces targeted, effective modifications that yield consistent performance gains. This suggests that task-specific fine-tuning can be more effective than using a general-purpose model of a much larger scale.

### 5.3 LEARNING PATTERN ANALYSIS

Figure 3a presents the learning curve of the RFTHGS framework, demonstrating stable and monotonic convergence. The average reward increases smoothly without significant oscillations, indicating a well-structured learning landscape with the effective design of our reward function. A critical inflexion occurs around step 200, where the generated operator surpasses the expert-designed baseline, marking the transition from learning executable operators to discovering superior heuristics. Beyond this intersection point, the curve continues to show consistent improvement, ultimately achieving substantially higher performance. This smooth progression shows that RFTHGS can effectively guide the LLM in generating increasingly sophisticated crossover operators. Figure 3b reveals the underlying learning patterns through the dynamics of the reward distribution. The heat map exhibits a clear curriculum learning pattern that precisely echoes our multi-tiered reward design. Initially, the density concentrates at lower rewards as the model masters syntactic correctness and compilability. Subsequently, the distribution shifts toward intermediate rewards, corresponding to the phase where operators become executable and yield valid solutions. Finally, the density center progressively migrates to the highest reward region, indicating the refinement toward operators that consistently outperform human-designed ones. This three-phase progression validates the effectiveness of our hierarchical reward structure in decomposing the complex operator design task into manageable learning stages.

### 5.4 GENERALIZATION PERFORMANCE ON ITERATIONS

The generalization capability of RFTHGS is further assessed across two joint dimensions, i.e., iteration count and problem size. Regarding iteration generalization, models trained with an 800-iteration budget (RFTHGS$_{800}$) maintain robust performance when evaluated at higher budgets of 1000 and 1200 iterations, consistently outperforming expert-designed baselines. This indicates that the optimized operators retain their efficacy beyond their training configuration. In terms of problem size, although trained exclusively on instances with $n < 400$, RFTHGS generalizes effectively to significantly larger problems (up to $n = 1000$). This joint generalization underscores the robustness and strong out-of-distribution scalability of our method. The results are shown in the last two rows (grey areas) of Table 1.

## 6 CONCLUSION

This paper introduces RFTHGS, a reinforcement learning framework that optimizes operators in the Hybrid Genetic Search (HGS) solver for solving the Capacitated Vehicle Routing Problem (CVRP). By fine-tuning with domain-specific rewards, we demonstrate that specialized small LLMs can surpass large general and deep thinking ones like GPT-4o, GPT-o4-mini, and GPT-o3 with trillions of parameters. Our core innovation is a novel RL-based fine-tuning paradigm guided by solution quality, featuring a multi-tiered reward mechanism with anti-plagiarism caching for progressive learning. Extensive experiments on CVRPLIB benchmarks confirm that the crossover (and subpopulation) operator generated by our method demonstrates superior performance over the expert-designed operator within the HGS framework, achieving substantial improvements, particularly on large-scale instances with up to 1,000 nodes. To our knowledge, this is the first work to show that a small, fine-tuned LLM can generate operators that exceed expert-crafted components in a leading combinatorial optimization solver. In future work, we plan to extend this framework to more other operators inside HGS and solve more types of VRPs.

## 7 ETHICS STATEMENT

This study involves no personal data, human subjects, or other sensitive content and therefore does not raise significant ethical concerns. The only potential risk lies in the fact that operators generated by LLMs may contain bugs which, if deployed without thorough validation, could cause losses.

## 8 REPRODUCIBILITY STATEMENT

To ensure the reproducibility of our work, we have provided comprehensive experimental details throughout this paper. Section 5.1 presents complete experimental configurations and environment specifications, while the Appendix A.3 includes detailed prompts. These materials provide sufficient information for independent reproduction of our experimental results. Furthermore, we open-source our full code base and model weights to further improve reproducibility: https://github.com/zaodushi/RFTHGS.

### ACKNOWLEDGMENTS AND DISCLOSURE OF FUNDING

This research is supported by the National Research Foundation, Singapore under the AI Singapore Programme (AISG Award No: AISG3-RPGV-2025-017).

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

# A APPENDIX

## A.1 ABLATION STUDIES ON REWARD DESIGN

To benchmark the effectiveness of our continuous reward design (Equation 3), we compare it with a discrete version defined as follows:

$$
r_d(o) = \begin{cases}
-1 & o \notin \mathrm{C} \\
-0.8 & o \in \mathrm{C}, o \notin \mathrm{E} \\
-0.9 & o \in \mathrm{C}, o \in \mathrm{E}, o \in \mathrm{P} \\
0 & o \in \mathrm{C}, o \in \mathrm{E}, o \notin \mathrm{P}, \phi_{\mathrm{HGS}}^{J}(o_{\mathrm{expert}}) < \phi_{\mathrm{HGS}}^{J}(o) \\
1 & o \in \mathrm{C}, o \in \mathrm{E}, o \notin \mathrm{P}, \phi_{\mathrm{HGS}}^{J}(o_{\mathrm{expert}}) > \phi_{\mathrm{HGS}}^{J}(o)
\end{cases}
\tag{4}
$$

Table 4: **Ablation study on reward design.** $\mathrm{FRTHGS}_d$ is the 14B LLM trained with reward in Equation 4. $\mathrm{FRTHGS}_c$ is the 14B LLM trained with reward in Equation 3. Shaded areas are generalization results.

| Methods | $n \in [100, 200)$ | | $n \in [200, 400)$ | | $n \in [400, 600)$ | | $n \in [600, 800)$ | | $n \in [800, 1000]$ | |
|---|---|---|---|---|---|---|---|---|---|---|
| | Gap% (↓) | Time (s) | Gap% (↓) | Time (s) | Gap% (↓) | Time (s) | Gap% (↓) | Time (s) | Gap% (↓) | Time (s) |
| HGS | **0.62** | 12.45 | 1.85 | 28.16 | 1.95 | 58.41 | 2.62 | 91.31 | 2.32 | 121.04 |
| $\mathrm{FRTHGS}_d$ | 0.83 | 11.75 | 1.78 | 28.89 | 1.92 | 58.12 | 2.64 | 94.30 | 2.30 | 120.28 |
| $\mathrm{FRTHGS}_c$ | 0.70 | 13.14 | **1.67** | 29.60 | **1.83** | 61.65 | **2.59** | 92.17 | **2.24** | 118.59 |

Table 4 demonstrates the clear advantage of our continuous reward design. $\mathrm{RFTHGS}_c$ consistently outperforms the discrete-reward variant $\mathrm{RFTHGS}_d$, particularly on larger problem sizes. The discrete reward's binary nature (0 or 1) provides limited guidance. Once an operator beats the baseline, the gradient vanishes as all improvements receive the same reward, and the advantage is thus 0 (Equation 2). In contrast, our continuous reward offers feedback proportional to performance gains, enabling sustained refinement and explaining its superior performance.

## A.2 THE USE OF LARGE LANGUAGE MODELS (LLMS)

We clarify that all intellectual contributions in this work, from initial idea conception and algorithm design to experimental implementation and result validation, were conducted exclusively by the human authors. While we employed LLMs during the manuscript preparation phase to refine language expression and improve readability, their role was strictly limited to linguistic polishing. The manuscript's structure, core arguments, and all substantive content were determined entirely by the human authors, with LLMs serving merely as an auxiliary tool for enhancing clarity and grammatical accuracy, similar to traditional proofreading services.

## A.3 PROMPT TEMPLATE

```
# ROLE: Expert C++ Optimization Engineer for Vehicle Routing Problems

You are a senior C++ optimization engineer with expertise in algorithmic
    optimization, particularly for Vehicle Routing Problems (VRP). Your
    task is to analyze and improve the selective_route_exchange.cpp file'
    s crossover algorithm.

## TASK OVERVIEW
You are given the file selective_route_exchange.cpp (full listing below).
Your goal is to make ONE small, reliable modification that tends to
    create
children with better penalised cost (Solution quality ↑) while keeping
    runtime
and interface intact.
```

```
## THINKING PROCESS REQUIREMENTS
1. First, thoroughly analyze the current implementation to understand:
   - The algorithm's purpose and workflow
   - Key decision points and heuristics
   - Performance bottlenecks or optimization opportunities
   - Any constraints that must be preserved

2. Generate at least 3 different modification approaches, evaluating each
     on:
   - Potential improvement to solution quality
   - Impact on runtime performance
   - Compatibility with existing code
   - Risk of introducing bugs or side effects

3. For your chosen modification:
   - Justify why it's likely to improve solution quality
   - Verify it maintains the function signature and behavior
   - Double-check for compatibility with the rest of the codebase
   - Consider edge cases and verify robustness

############################################################
## HARD RULES (Mandatory Verification Checklist)

1. □ Keep the function signature and namespace exactly the same:
      pyvrp::crossover::selectiveRouteExchange(...)

2. □ The file must still compile under C++17 with the current #include
      lines.
      You may NOT remove #include directives.

3. □ Do not change any public headers, class interfaces, or external
      behaviour
      except for the improved offspring quality.

4. □ DO NOT fabricate or use non-existent or unmentioned attributes or
      methods.
      Verify every method you use exists in the provided code or
         documentation.

5. □ Wrap the code you output with ```cpp and ```.

6. □ Mark ALL your modifications with clear "// MODIFY: XXX" comments
      explaining the change.

7. □ You must make at least one modification; DO NOT copy the original
      code.

8. □ Before finalizing, double-check that your modification:
   - Does not introduce new parameters
   - Does not change the function's contract
   - Is focused on improving solution quality, not runtime
   - Is fully compatible with the existing codebase
   - Uses only documented methods and attributes

############################################################
## DELIVERABLES (strict):

A. ≤ 2-sentence summary of the optimization idea, clearly explaining how
     it improves solution quality.

B. Output the FULL C++ code with your modifications. Mark all changes
     with "// MODIFY: XXX" comments.

C. Brief explanation of your verification process and why you're
     confident the modification will:
```

    - Improve solution quality
    - Maintain compatibility with the existing codebase
    - Not significantly impact runtime performance

```
############################################################
## SCORING AND EVALUATION
```

We will benchmark on a fixed random seed over several CVRP instances.
Your patch should reduce the average optimal gap in ≥90% of the instances without
increasing total runtime by >3%.

Key considerations for high-quality solutions:
- More efficient route structures (fewer vehicles, shorter routes)
- Better client assignment to routes based on spatial relationships
- Improved handling of capacity constraints
- Preservation of high-quality route segments during crossover
- Better diversity in the generated offspring

```
############################################################
```

## selective_route_exchange.cpp
```cpp
{code}
```

## Extra Information:
## DOMAIN KNOWLEDGE: CVRP AND CROSSOVER OPERATIONS

The Selective Route Exchange is a crossover operation for the Capacitated Vehicle Routing Problem (CVRP). The algorithm:
1. Selects routes from two parent solutions
2. Exchanges these routes to create offspring
3. Aims to preserve beneficial route structures while creating new combinations

### Key Optimization Areas to Consider:
- Route selection strategy (which routes to exchange)
- Client-to-route assignment decisions
- Proximity/distance calculations between routes or clients
- Handling of capacity constraints
- Diversity generation in offspring solutions

## Essential Fields and Methods for CVRP Crossover

**ProblemData Key Methods:**
- `numLocations()` – Returns `size_t` total number of locations (depots + clients)
- `numClients()` – Returns `size_t` number of client locations
- `centroid()` – Returns `std::pair<double, double>` center of all client locations
- `client(idx)` – Returns `ProblemData::Client` with coordinates (x, y)

**Route Key Methods:**
- `centroid()` – Returns `std::pair<double, double>` center of route's client locations
- `vehicleType()` – Returns `VehicleType` (size_t) vehicle type index
- `begin()` / `end()` – Iterator support for visiting clients in route
- `size()` – Returns `size_t` number of clients in route
- `visits()` – Returns `std::vector<Client>` all client indices in route order

**Route Construction:**
- `Route(data, visits, vehicleType)` – Constructor taking `std::vector<Client>` visits and vehicle type

```
**Client Iteration:**
- Routes are iterable containers of `Client` (size_t) indices
- Use range-based for loops: `for (Client c : route)` to access all
    clients in route
- Client coordinates: `data.client(c).x`, `data.client(c).y`
```

## A.4 RFTHGS PSEUDO CODE

---

**Algorithm 1:** RFTHGS: Reinforcement Finetuning For Refining HGS

---

**Input:** Initial policy $\pi_{\theta_{old}}$, full instance set $\mathcal{I}$, instance batch size $B$, clipping parameters
       $\varepsilon_{lower}, \varepsilon_{upper}$, group size $G$

**Output:** Optimized policy $\pi_\theta$

Initialize $\theta \leftarrow \theta_{old}$;

**foreach** *iteration* $1, 2, \ldots, N$ **do**
    Draw a random subset $\mathcal{B} \subset \mathcal{I}$ with $|\mathcal{B}| = B$;
    **foreach** *instance* $I \in \mathcal{B}$ **do**
        `// Step 1: Generate operators with LLM`
        Construct prompt $q$ for CrossOver operator optimization;
        Sample $G$ operators $\{o_1, \ldots, o_G\} \sim \pi_{\theta_{old}}(\cdot \mid q)$;
        `// Step 2: Evaluate with PyVRP`
        **foreach** *operator* $o_i$ **do**
            Run PyVRP solver with $o_i$ substituted on instance set $\mathcal{B}$;
            Obtain objective value and compute reward $r_i$;
        **end**
        `// Step 3: Compute advantages`
        $\mathbf{r} = (r_1, r_2, \ldots, r_G)$
        Normalised reward for each operator: $\hat{A}_{i,t} = \dfrac{r_i - \mathrm{mean}(\mathbf{r})}{\mathrm{std}(\mathbf{r})}$;
        `// Step 4: Update policy with DAPO`
        **foreach** *token position* $t$ **do**
            $r_t(\theta) \leftarrow \dfrac{\pi_\theta(o_t \mid q, o_{<t})}{\pi_{\theta_{old}}(o_t \mid q, o_{<t})}$;
            $L_{policy} \leftarrow \mathbb{E}\big[\min\big(r_t(\theta)\,\hat{A}_t,\; \mathrm{clip}\big(r_t(\theta), 1 - \varepsilon_{lower}, 1 + \varepsilon_{upper}\big)\,\hat{A}_t\big)\big]$;
        **end**
        $\theta \leftarrow \theta + \alpha \nabla_\theta L_{policy}$;
    **end**
**end**

---

## A.5 OPTIMIZING OTHER OPERATORS USING RFTHGS

The RFTHGS framework is not only suitable for optimizing the crossover operator but can also be applied to optimize other operators (or modules) influencing solution quality within HGS. In this section, we demonstrate the application of RFTHGS to optimize the `subpopulation (subp)` operator in HGS. The primary function of this operator is to manage population control by determining which individuals to eliminate based on cost and diversity metrics, ensuring the population size remains within defined thresholds.

To rigorously test the generalizability of our framework, we applied RFTHGS to the Capacitated Vehicle Routing Problem with Time Windows (CVRPTW) variant. It is important to emphasize that the configuration of the RFTHGS framework remained consistent with the experiments described in the main manuscript, where the POMDP formulation, training algorithm, base model, and reward calculation were identical.

**Experimental Setup** For training, we generate a dataset comprising 50 randomly generated CVRPTW instances with problem sizes ranging from 100 to 300 customers. For the testing phase, we generated 100 new random instances for each problem size, extending up to 1000 customers.

**Results and Analysis**  The performance of the RFTHGS-optimized `subp` operator on CVRPTW instances is summarized in Table 5, where the relative gap to the original HGS, calculated as $(Cost_{RFTHGS} - Cost_{HGS})/Cost_{HGS}$, is reported. A negative value indicates that the RFTHGS-optimized operator outperforms the baseline HGS.

Table 5: Performance of the optimized subpopulation operator on CVRPTW instances. Values represent the relative gap (%) compared to the baseline HGS.

| instance size | 100 | 200 | 300 | 500 | 1000 |
|---|---|---|---|---|---|
| HGS-subp (%) | 0.000 | 0.000 | 0.000 | 0.000 | 0.000 |
| RFTHGS-subp (%) | -0.088 | -0.213 | -0.299 | -0.289 | -0.224 |

As shown in Table 5, the optimized `subp` operator consistently outperforms the standard HGS across all problem sizes. Furthermore, the operator demonstrates strong generalization capabilities, maintaining superior performance on much larger problem sizes (500 and 1000) that were not seen during training, achieving relative gaps of -0.289% and -0.224%, respectively.

**Mechanism of Improvement**  An analysis of the code generated by RFTHGS reveals that the enhanced `subp` operator modifies the fitness evaluation logic. The LLM introduced a third ranking metric based on the *number of routes* in a solution. This new metric is weighted and combined with the existing cost and diversity rankings to determine individual survival during parent selection.

This modification proves effective because it introduces a selective pressure that favors solutions utilizing fewer vehicles. By biasing the population towards more compact configurations, the algorithm indirectly promotes lower total costs, as fewer routes inherently reduce the cumulative distance associated with frequent returns to the depot. These results confirm that RFTHGS is a framework capable of optimizing diverse components within mature solvers across different problem variants.

## A.6  TRAINING COST ANALYSIS

We provide a breakdown of the computational resources for the RL training of our 14B model. The total training duration was approximately 98 GPU hours per device. Analyzing the cost per training step reveals a total duration of roughly 660 seconds, which is composed of 360 seconds for LLM training (inference, backward propagation, and updates) and 300 seconds for operator evaluation (execution and reward calculation).

Since reward computation constitutes a significant portion of the training cycle, future implementations can mitigate this bottleneck by adopting fully asynchronous training paradigms and utilizing lightweight sandbox environments, such as `llm-sandbox` (vndee, 2025). These tools enable the isolation of code execution on CPU-only nodes, allowing for massive parallelization and reduced GPU idle time.

## A.7  AST-BASED ANTI-PLAGIARISM IMPLEMENTATION DETAILS

This mechanism determines whether two LLM-generated C++ codes are structurally equivalent by comparing their Abstract Syntax Trees (ASTs) after normalization.

The process begins by invoking the Clang compiler frontend as a subprocess on each source file. Clang is called with the *-ast-dump=json* flag, which outputs the full AST in JSON format.

Once the raw AST is obtained, we traverses the tree depth-first, locating all function definitions that have an actual body. For each function, it constructs a simplified dictionary retaining only the function name, type signature, parameter list, and body, discarding all metadata such as source locations, node IDs, and mangled names.

The function body is then recursively simplified. After simplification, the functions undergo alpha-renaming. This ensures that two programs with identical structure but different identifier choices produce the same normalized AST. Finally, the two normalized ASTs are serialized to JSON with deterministic key ordering and compared via string equality.

This mechanism helps the framework determine whether the LLM has made genuine changes to the code, helping to prevent reward hacking.

## A.8 COMPARISON WITH COMPASS

We compare with COMPASS Chalumeau et al. (2023), another SOTA neural combinatorial optimization (NCO) baseline, published at NeurIPS 2023. Our implementation ran on CUDA 12.8 while the original COMPASS was developed for CUDA 11, which may lead to a performance gap relative to the reported results; we did not find critical bugs in our reproduction.

Table 6 reports results on CVRPLIB X instances (sizes 100–200). Our method maintains superioir performance compared to the COMPASS baseline.

Table 6: RFTHGS vs. COMPASS on CVRPLIB X instances.

| Instance | Obj. COMPASS | Gap COMPASS (%) | Obj. RFTHGS | Gap RFTHGS (%) | Optimal |
|---|---|---|---|---|---|
| X-n101-k25 | 42674 | 54.67 | 27597 | 0.02 | 27591 |
| X-n106-k14 | 34259 | 29.95 | 26546 | 0.70 | 26362 |
| X-n110-k13 | 15310 | 2.27 | 15084 | 0.75 | 14971 |
| X-n115-k10 | 16544 | 29.79 | 12747 | 0.00 | 12747 |
| X-n120-k6 | 14897 | 11.74 | 13392 | 0.45 | 13332 |
| X-n125-k30 | 71921 | 29.50 | 55825 | 0.51 | 55539 |
| X-n129-k18 | 29578 | 2.20 | 29105 | 0.57 | 28940 |
| X-n134-k13 | 23375 | 114.13 | 11050 | 1.23 | 10916 |
| X-n139-k10 | 15366 | 13.07 | 13612 | 0.16 | 13590 |
| X-n143-k7 | 52516 | 234.50 | 15743 | 0.27 | 15700 |
| X-n148-k46 | 47049 | 8.29 | 44036 | 1.35 | 43448 |
| X-n153-k22 | 28583 | 34.70 | 21337 | 0.55 | 21220 |
| X-n157-k13 | 18697 | 10.79 | 16925 | 0.29 | 16876 |
| X-n162-k11 | 44795 | 216.84 | 14175 | 0.26 | 14138 |
| X-n167-k10 | 24145 | 17.46 | 20894 | 1.64 | 20557 |
| X-n172-k51 | 63857 | 40.02 | 45939 | 0.73 | 45607 |
| X-n176-k26 | 59699 | 24.86 | 48370 | 1.17 | 47812 |
| X-n181-k23 | 28537 | 11.61 | 25734 | 0.65 | 25569 |
| X-n186-k15 | 53322 | 120.84 | 24299 | 0.64 | 24145 |
| X-n190-k8 | 19590 | 15.37 | 17294 | 1.85 | 16980 |
| X-n195-k51 | 73445 | 66.07 | 44681 | 1.03 | 44225 |
| X-n200-k36 | 85408 | 45.80 | 58862 | 0.48 | 58578 |
| Average | 39253.05 | 51.57 | 27420.32 | 0.70 | 27220.14 |

