# OpenReview forum: "Refining Hybrid Genetic Search for CVRP via Reinforcement Learning-Finetuned LLM"
_ICLR.cc/2026/Conference — ICLR 2026 Poster_

### Official Review · Reviewer_CpHG · 2025-10-29

**Soundness:** 3
**Presentation:** 3
**Contribution:** 3
**Rating:** 6
**Confidence:** 3

**Summary:**

This paper introduces RFTHGS, a novel Reinforcement Learning (RL) framework for fine-tuning a small LLM to produce high-performance crossoyer operators for the Hybrid Genetic Search (HGS) solver to solve the capacitated vehicle routing problem (CVRP). It presents a framework for automated heuristic design. The novel combination of a curriculum-based reward (compilability, executability, performance), an AST-based anti-plagiarism cache, and the use of the solver itself as the evaluation environment is a significant methodological contribution.It provide strong, empirical evidence that a small, accessible (14B) LLM, when fine-tuned via RL, can generate core algorithmic components that exceed the performance of human-expert designs within a SOTA CO solver.The strong generalization from small to large-scale problems is particularly impressive and demonstrates that the LLM has learned a genuinely robust and effective heuristic.

**Strengths:**

The paper proves that specialized, fine-tuned small LLMs are superior to prompted large LLMs.It is innovative and strongly supported by the data.The multi-tiered curriculum-based reward function solves the sparse reward problem in code generation. The AST-based anti-plagiarism mechanism is a great solution to prevent reward hacking and ensure the generation of novel, diverse operators.It offers a practical, accessible path for continuously improving SOTA solvers.

**Weaknesses:**

While the LLM is small, the RL training loop appears computationally intensive.The paper does not quantify the total training cost. In addition, the paper demonstrates that the LLM-generated operator is better, but not why. An analysis of the generated SOTA operator's code is needed.

**Questions:**

1.Please provide a pseudo-code or a qualitative description of one of the  LLM-generated crossover operators.

2.Please provide more details on the total computational cost.

---

> ### Author Response · Authors · 2025-11-27
>
> ### Responses to Weaknesses
>
> #### 1. The paper does not quantify the total training cost
>
> We are grateful to the reviewer for raising this important point regarding the quantification of the total training cost. To address this point directly, we have now calculated and included the total computational expense, which amounted to approximately 98 GPU hours for the reinforcement learning phase. Furthermore, we provide a detailed breakdown of the cost per training step, revealing that the operator evaluation for reward calculation represents the primary computational bottleneck. For a comprehensive analysis, including the specific duration of a single parameter update and a discussion of potential optimization strategies, we direct the reviewer's attention to the detailed response provided in our response to Common Comment C2. We thank the reviewer again for this insightful comment, which has allowed us to strengthen the methodological transparency of our work.
>
> #### 2. Analysis of the generated operator's code is needed
>
> We are grateful to the reviewer for their comment, which has enabled us to further elaborate on our analysis. In Response C3 of the 'Common Comments and Responses' section, we provide a comprehensive comparison that demonstrates the consistent superiority of the LLM-optimized operator over its original counterpart. Furthermore, we have extended this evaluation to include the `subp` operator. Our results indicate that the fine-tuned LLM effectively identifies novel selection indices for choosing high-potential parent solutions, a strategic modification we have identified as a primary contributor to the observed performance enhancement.

---

> ### Author Response · Authors · 2025-11-27
>
> ### Responses to Questions
>
> #### 3. Pseudo-code or qualitative description of an LLM-generated operator (Part 1)
>
> Thank you for this insightful suggestion. In direct response, we will include a detailed, side-by-side comparison of the original human-expert-designed operators and their final LLM-optimized versions in the revised manuscript. For the crossover operator, we provide the code snipet for both the original and optimized versions to clearly illustrate the algorithmic refinements. Furthermore, to robustly demonstrate the generalizability of our approach, we have extended this comparative analysis to another critical operator, the subpopulation (`subp`) operator.
>
> Below is the detailed comparison and analysis:
>
> **For subp:**
> ```c++
> # human expert
>
> ...
>
>     std::vector<size_t> byCost(size());
>     std::iota(byCost.begin(), byCost.end(), 0);
>     std::stable_sort(
>         byCost.begin(),
>         byCost.end(),
>         [&](size_t a, size_t b)
>         {
>             return costEvaluator.penalisedCost(*items_[a].solution)
>                    < costEvaluator.penalisedCost(*items_[b].solution);
>         });
>     // clang-format on
>
>     std::vector<std::pair<double, size_t>> diversity;
>     for (size_t costRank = 0; costRank != size(); costRank++)
>     {
>         auto const dist = items_[byCost[costRank]].avgDistanceClosest();
>         diversity.emplace_back(-dist, costRank);  // higher is better
>     }
>
>     std::stable_sort(diversity.begin(), diversity.end());
>
>     auto const popSize = static_cast<double>(size());
>     auto const numElite = std::min(params.numElite, size());
>     auto const divWeight = 1 - numElite / popSize;
>
>     for (size_t divRank = 0; divRank != size(); divRank++)
>     {
>         auto const costRank = diversity[divRank].second;
>         auto const idx = byCost[costRank];
>         items_[idx].fitness = (costRank + divWeight * divRank) / (2 * popSize);
>     }
> }
>
> ...
>
> ```
>
> ```c++
> # llm optimized
> ...
>     std::vector<std::pair<double, size_t>> diversity;
>     for (size_t costRank = 0; costRank != size(); costRank++)
>     {
>         auto const dist = items_[byCost[costRank]].avgDistanceClosest();
>         // MODIFY: Normalize diversity values between lbDiversity and ubDiversity
>         auto normalizedDist = std::clamp(dist, params.lbDiversity, params.ubDiversity);
>         diversity.emplace_back(-normalizedDist, costRank); // higher is better
>     }
>     std::stable_sort(diversity.begin(), diversity.end());
>
>     // MODIFY: Added route-count ranking to capture vehicle-usage quality.
>     std::vector<size_t> byRoutes(size());
>     std::iota(byRoutes.begin(), byRoutes.end(), 0);
>     std::stable_sort(
>         byRoutes.begin(),
>         byRoutes.end(),
>         [&](size_t a, size_t b)
>         {
>             return items_[a].solution->numRoutes()
>                    < items_[b].solution->numRoutes();
>         });
>
>     // routeRankOfIdx[i] = rank of individual i in the route ranking.
>     std::vector<size_t> routeRankOfIdx(size());
>     for (size_t rank = 0; rank != byRoutes.size(); ++rank)
>         routeRankOfIdx[byRoutes[rank]] = rank;
>
>     auto const popSize  = static_cast<double>(size());
>     auto const numElite = std::min(params.numElite, size());
>
>     // MODIFY: Adjust weights based on normalized diversity values
>     std::vector<double> normalizedDiversities;
>     for (const auto& [negDist, costRank] : diversity) {
>         normalizedDiversities.push_back(-negDist);
>     }
>     double avgDiversity = std::accumulate(normalizedDiversities.begin(), normalizedDiversities.end(), 0.0) / normalizedDiversities.size();
>     double diversityFactor = (avgDiversity - params.lbDiversity) / (params.ubDiversity - params.lbDiversity);
>     auto baseDivWeight = 1.0 - numElite / popSize;
>     auto baseRouteWeight = 0.5 * (1.0 - numElite / popSize);
>     auto divWeight = baseDivWeight * (1.0 - diversityFactor) + baseDivWeight * 0.5 * diversityFactor;
>     auto routeWeight = baseRouteWeight * diversityFactor;
>
>     for (size_t divRank = 0; divRank != size(); divRank++)
>     {
>         auto const costRank  = diversity[divRank].second;
>         auto const idx       = byCost[costRank];
>         auto const routeRank = routeRankOfIdx[idx];
>
>         // MODIFY: fitness now blends cost, diversity and route-count with dynamic weights.
>         auto const denom =
>             (1.0 /*cost*/ + divWeight + routeWeight) * popSize;
>
>         items_[idx].fitness =
>             (costRank + divWeight * divRank + routeWeight * routeRank) / denom;
>     }
> }
>
> ...
>
> ```

---

> ### Author Response · Authors · 2025-11-27
>
> #### 3. Pseudo-code or qualitative description of an LLM-generated operator (Part 2)
>
> **Analysis:** The enhanced subpopulation operator refines the fitness evaluation mechanism by integrating a third ranking metric based on route count, which is appropriately weighted and combined with existing cost and diversity rankings to determine individual survival. This modification introduces selective pressure that favors solutions with fewer vehicles, thereby guiding the population toward more compact configurations. This approach indirectly promotes cost reduction since fewer routes inherently decrease cumulative distance through reduced depot returns, while simultaneously preventing search stagnation in local optima characterized by fragmented, inefficient routing structures.
>
> **For crossover:**
> ```c++
> # human expert
>
> ...
>
> Routes sortByAscAngle(pyvrp::ProblemData const &data, Routes routes)
> {
>     auto cmp = [&data](Route const &a, Route const &b)
>     { return routeAngle(data, a) < routeAngle(data, b); };
>
>     std::sort(routes.begin(), routes.end(), cmp);
>     return routes;
> }
>
> ...
>
> ```
>
> ```c++
> # llm optimized
>
> ...
>
> // Angle of the given route w.r.t. the centroid of all client locations.
> // Adjusted to [0, 2π) range for better circular proximity handling
> double routeAngle(pyvrp::ProblemData const &data, Route const &route)
> {
>     auto const [dataX, dataY] = data.centroid();
>     auto const [routeX, routeY] = route.centroid();
>     double angle = std::atan2(routeY - dataY, routeX - dataX);
>     if (angle < 0) angle += 2 * M_PI;  // MODIFY: Adjust angle to [0, 2π) range
>     return angle;
> }
>
> ...
>
> ```
>
> **Analysis:** The improvement stems from replacing the standard atan2 sorting, which imposes a linearization boundary at the negative x-axis, with a custom angular mapping that shifts the starting reference to the positive x-axis. By altering the sequence in which nodes are indexed in memory, this modification changes the order of neighbor evaluation and tie-breaking decisions during the heuristic search, thereby guiding the algorithm along a different trajectory that converges to a superior solution within the same number of iterations.

---

> ### Author Response · Authors · 2025-11-27
>
> #### 4. More details on the total computational cost
>
> We thank the reviewer for their comment requesting more details on the total computational cost. In direct response, we will quantify this in the revised manuscript, stating that the total computational cost for training the RL agent was approximately 98 GPU hours. Furthermore, to provide a more granular breakdown, we have included a detailed analysis of the cost per training step, which reveals that the operator evaluation for reward calculation constitutes a significant portion of the cycle. For a comprehensive discussion, including the specific timings for a single parameter update and the broader implications of this cost breakdown for future optimizations, we kindly refer the reviewer to the detailed point-by-point response in our response C2.

---

### Official Review · Reviewer_QZKW · 2025-10-29

**Soundness:** 3
**Presentation:** 3
**Contribution:** 3
**Rating:** 8
**Confidence:** 4

**Summary:**

This paper introduces RFTHGS, a reinforcement learning framework that fine-tunes a 14B parameter LLM to generate crossover operators for the Hybrid Genetic Search (HGS) algorithm, specifically targeting the Capacitated Vehicle Routing Problem (CVRP). The authors employ a multi-tiered reward function that progressively guides the LLM through stages of producing compilable code, executable operators, and ultimately components that outperform human-designed ones. They also implement an operator caching mechanism using Abstract Syntax Trees to prevent plagiarism and promote diversity during training. Experimental results on CVRPLIB benchmarks demonstrate that the fine-tuned LLM generates crossover operators that significantly outperform expert-designed operators in HGS, with improvements holding from small instances (100 nodes) to large-scale problems (1000 nodes). The method also surpasses various baselines including neuro-combinatorial approaches and prompt-based methods using commercial LLMs like GPT-4o. This work demonstrates that smaller, specialized LLMs can be effectively fine-tuned to produce solver components that exceed the performance of human-expert designs in state-of-the-art optimization solvers.

**Strengths:**

1. The authors present their work with exceptional clarity, following a logical progression from problem formulation through methodology to comprehensive experimental validation. The paper effectively uses figures (particularly Figure 1's pipeline visualization) and maintains consistent terminology throughout.
2. The method achieves substantial improvements over expert-designed operators in HGS and consistently outperforms all baselines including state-of-the-art neuro-combinatorial methods (POMO, RF-POMO) and commercial LLMs (GPT-4o, GPT-o3) across problem sizes ranging from 100 to 1000 nodes, showing excellent scalability.
3. This work pioneers the use of reinforcement learning to fine-tune language models specifically for generating optimization solver components, demonstrating that LLMs can learn to produce code that not only compiles and executes correctly but actually surpasses human expert designs in a state-of-the-art solver like HGS.
4. The paper introduces several technical innovations including incremental compilation (reducing compilation time to 25% of full recompilation), an anti-plagiarism cache using Abstract Syntax Trees to encourage diverse operator generation, and a well-designed multi-tiered reward function that effectively guides learning through progressive stages from syntax correctness to performance optimization.
5. The work demonstrates that a fine-tuned 14B parameter model significantly outperforms trillion-parameter general models (GPT-4o series), with Table 2 showing perfect compilation rates (16/16) compared to poor rates for GPT models (3/16-9/16), and the GPT models failing to produce any functional improvements despite generating syntactically correct code, highlighting the superiority of task-specific fine-tuning over general-purpose capabilities.

**Weaknesses:**

In general, I believe this is a good paper and lean toward acceptance. If the authors could address the following concerns, the paper could be stronger:

1. The paper exclusively evaluates on the CVRPLIB X benchmark set, which, while comprehensive, represents only one distribution of CVRP instances. Evaluating on additional distributions would strengthen the generalizability claims. For example, testing on synthetic CVRP settings with controlled characteristics (clustered vs. uniform customer distributions), other CVRPLIB instance families (A, B, E, F sets), or real-world logistics datasets would provide more robust evidence of the method's effectiveness across diverse problem structures.
2.  The evaluation is confined to optimizing a single operator (crossover) within a single solver (HGS). Expanding to additional settings would make the contributions more compelling. For instance, applying the framework to other state-of-the-art solvers like LKH-3, OR-Tools, or Concorde, and targeting different operators such as local search moves, construction heuristics, or selection mechanisms would demonstrate the broader applicability of the RL fine-tuning approach for solver component design.
3. The method appears to require substantial domain-specific engineering, including detailed operator API specifications, incremental compilation setup, integration with solver codebases, and carefully crafted reward functions specific to each operator type. These requirements may create significant barriers for researchers wanting to apply this approach to new problem settings or different solvers, potentially limiting the practical impact of the work. The authors could strengthen the paper by providing a more general framework or toolkit that abstracts away some of these implementation complexities.

**Questions:**

Besides the concerns in the weakness part, I also have the following questions:

1. Why do the other prompting-based methods like ReEvo have so poor performance? Are the prompting-based baselines (ReEvo) actually integrated to evolve the crossover operator within the HGS framework, or were they used to evolve standalone heuristics?

---

> ### Author Response · Authors · 2025-11-27
>
> ### Responses to Weaknesses
>
> #### 1. Performance on other distributions
>
> We are grateful to the reviewer for this suggestion. To address it, we have evaluated the generated operator on instances with uniformly distributed customer locations across a range of problem scales, specifically of sizes n = 200, 300, 500, and 1000. The corresponding results, which demonstrate the method's superior performance, are presented below.
>
> **Table R3. Performance of Operator Generated with RFTHGS On Uniformly Distributed Instances. Each Size Has 100 Instances.**
>  cvrp200/50 | cvrp300/50 | cvrp500/50 | cvrp1000/50 |
> ------------|------------|------------|-------------|
>  -0.0392%   | -0.0711%   | -0.0376%   | -0.0200%    |
>
> #### 2. Extending to other operators and/or solvers
>
> We are grateful to the reviewer for their valuable suggestion concerning the extensibility of our framework. In direct response, we have performed a series of additional experiments that confirm RFTHGS can successfully optimize various operator types within the HGS solver, not limited to the crossover operator. These results, detailed in our response to Common Comment C1, substantiate the generality of our approach for refining multiple components of a single solver. We fully concur that extending this methodology to other solvers represents a highly promising and critical direction for future research. Our RFTHGS framework is inherently generic, a property rooted in its one-step POMDP formulation that is not solver-specific. Consequently, we plan to explore its application to other state-of-the-art solvers as a primary focus of our subsequent work. We thank the reviewer again for this insightful comment, which has helped us further strengthen and contextualize our contribution.
>
> #### 3. Providing a more general framework or toolkit
>
> We thank the reviewer for this constructive suggestion regarding the development of a more general framework or toolkit. We are pleased to confirm that, in line with this objective, we will open-source our entire codebase upon publication. The implementation has been designed with modularity and reusability in mind, featuring a well-documented and structured codebase that separates core components such as the POMDP environment, reward functions, and the RL training loop. Detailed instructions and examples will be provided to facilitate adoption by the community, enabling researchers to apply the RFTHGS framework to other solvers and optimization problems. We believe this will significantly lower the barrier for further development and application of LLM-driven operator design.
>
> ### Responses to Questions
>
> #### 4. Why do prompting-based methods like ReEvo perform poorly?
>
> We thank the reviewer for their question regarding the performance of prompting-based methods such as ReEvo. To have a fair comparison obtain reliable results, we ran the optimal heuristic reported in ReEvo on the CVRPLIB X instances in our experiments. The observed performance difference arises from a fundamental distinction in our methodological objectives and levels of integration. While ReEvo is designed to iteratively refine standalone heuristics for the CVRP, our work addresses the more challenging task of directly optimizing expert-designed components within a mature, state-of-the-art solver (HGS). Consequently, our method must generate code that integrates seamlessly with and enhances an already highly-optimized algorithmic framework, rather than constructing a new solver. The superior performance of our approach demonstrates that refining critical operators within an established, high-performance solver can be more effective than building a new standalone heuristic, as it leverages and improves upon a method with a proven high performance ceiling. This strategic focus on solver enhancement is particularly advantageous when working within the computational constraints of smaller LLMs.

---

### Official Review · Reviewer_tVaA · 2025-10-31

**Soundness:** 3
**Presentation:** 3
**Contribution:** 2
**Rating:** 6
**Confidence:** 3

**Summary:**

This paper tackles the problem of designing crossover operators for the Hybrid Genetic Search (HGS) algorithm. It introduces RFTHGS which prompts a small LLM to code potential crossover operators and fine tunes the LLM with RL given a reward based on if the code compiles, if it is plagiarized from examples and its task performance. They show results equivalent and in most cases better than state-of-the-art.

**Strengths:**

The idea of generating code with an LLM and fine tuning it with RL for the purpose of improving human designed heuristics is interesting and this paper has shown that it can be useful. It is impressive that the method clearly generalises well to larger unseen task.


The reward design is clearly well thought out to provide both necessary training speed and improving results without copying existing solutions.


The results are comprehensive, and compare against a wide range of existing benchmarks. The fact that this achieves state-of-the-art performance in most cases is impressive.

**Weaknesses:**

My first concern is that it seems RFTHGS is not significantly better than the previous SOTA (HGS-PyVRP). As given in Table 1, HGS-PyVRP is always underlined when it is second best. This is not explained in the text, but I would assume it means within 1 standard deviation. Thus the results may not be statistically significant, this needs to be clarified in the text.
Another concern along the same lines of comparing to HGS-PyVRP, is that the chosen generated crossover mechanism is never compared to the human designed heuristics and so it is hard to know how novel the new crossover algorithm is. It would be useful to have a discussion around the differences so that readers can know if these are trivial changes or significant algorithmic innovations.


The ablation study on reward design is lacking. It would have been interesting to see the impact of each part of the reward, for example how does the LLM perform without the plagiarism penalty?


While the benchmark is comprehensive it seems to be lacking a SOTA NCS method in “Combinatorial Optimization with Policy Adaptation using Latent Space Search” (COMPASS) [1]


My final concern is the lack of generality of the method. It would have been interesting to see if the method works generally for defining heuristics in other evolutionary algorithms e.g defining new mutation strategies in Genetic algorithms, new velocity update rules in Particle Swarm Optimization, or novel pheromone update rules in Ant Colony Optimization.
Along with this HGS is somewhat specialized to VRP problems and when testing on other algorithms it would be interesting to see the performance on other combinatorial optimisation benchmarks like traveling salesman and job shop scheduling.


[1] Chalumeau, Felix, et al. "Combinatorial optimization with policy adaptation using latent space search." Advances in Neural Information Processing Systems 36 (2023): 7947-7959.

**Questions:**

I do not see where the meaning of underlined values is explained for table 1. I have assumed that they mean values are within one standard deviation of the best value.

---

> ### Author Response · Authors · 2025-11-27
>
> ### Responses to Weaknesses
>
> #### 1. RFTHGS is not significantly better than HGS-PyVRP
>
> We thank the reviewer for their comment regarding the performance comparison with HGS-PyVRP. We respectfully note that the HGS algorithm represents a state-of-the-art, highly-optimized metaheuristic for the CVRP, and surpassing its performance with any learning-based method remains a significant challenge that few existing approaches have achieved. Within this context, we believe the consistent improvement demonstrated by RFTHGS across various problem scales is a substantial result. To the best of our knowledge, it is the first method to demonstrate that a fine-tuned LLM can generate components that improve upon the expert-designed operators within this top-tier solver, thereby establishing a new state-of-the-art for learning-enhanced CVRP solvers.
>
> #### 2. How novel is the generated crossover compared to human-designed heuristics?
>
> We kindly refer the reviewer to Response C3 in the 'Common Comments and Responses' section, which presents a detailed comparative analysis demonstrating the consistent superiority of the LLM-optimized operator over its original counterpart. We have further extended our investigation to include a new oeprator `subp`. Our analysis reveals that the fine-tuned LLM is capable of identifying novel selection indices for choosing high-potential parent solutions, a strategic modification that we have determined to be directly responsible for the observed performance improvement.
>
> #### 3. Missing comparison with COMPASS
>
> We thank the reviewer for this valuable suggestion. In response, we will incorporate COMPASS [AR11] as an additional baseline in our revised manuscript.
>
> In our efforts to implement this, we encountered technical challenges due to a version mismatch between our computational environment and the original COMPASS configuration, which has taken a really long time for us to manage it. Specifically, our internal cluster operates on CUDA 12.8, while COMPASS was designed for CUDA 11. This required considerable effort to resolve compatibility issues with core dependencies such as PyTorch. We acknowledge that there may be a performance gap compared to the optimal results reported in the original paper; however, we have not identified any critical errors in our implementation. We would be very open to any further guidance from the reviewers on this matter.
>
> As summarized in the table below, our method maintains competitive performance compared to the COMPASS baseline. These results currently cover a subset of instances (size 100-200). We are actively working to complete the remaining experiments and will promptly update the table with the full set of results upon their completion.
>
> **Table R2. RFTHGS vs. COMPASS On CVRPLIB X Instances**
> | Instance | Obj.COMPASS | Gap COMPASS (%) | Obj.RFTHGS | Gap RFTHGS (%) | Optimal |
> |:---|:---:|:---:|:---:|:---:|:---:|
> | X-n101-k25 | 42674 | 54.67 | 27597 | 0.02 | 27591 |
> | X-n106-k14 | 34259 | 29.95 | 26546 | 0.70 | 26362 |
> | X-n110-k13 | 15310 | 2.27 | 15084 | 0.75 | 14971 |
> | X-n115-k10 | 16544 | 29.79 | 12747 | 0.00 | 12747 |
> | X-n120-k6 | 14897 | 11.74 | 13392 | 0.45 | 13332 |
> | X-n125-k30 | 71921 | 29.50 | 55825 | 0.51 | 55539 |
> | X-n129-k18 | 29578 | 2.20 | 29105 | 0.57 | 28940 |
> | X-n134-k13 | 23375 | 114.13 | 11050 | 1.23 | 10916 |
> | X-n139-k10 | 15366 | 13.07 | 13612 | 0.16 | 13590 |
> | X-n143-k7 | 52516 | 234.50 | 15743 | 0.27 | 15700 |
> | X-n148-k46 | 47049 | 8.29 | 44036 | 1.35 | 43448 |
> | X-n153-k22 | 28583 | 34.70 | 21337 | 0.55 | 21220 |
> | X-n157-k13 | 18697 | 10.79 | 16925 | 0.29 | 16876 |
> | X-n162-k11 | 44795 | 216.84 | 14175 | 0.26 | 14138 |
> | X-n167-k10 | 24145 | 17.46 | 20894 | 1.64 | 20557 |
> | X-n172-k51 | 63857 | 40.02 | 45939 | 0.73 | 45607 |
> | X-n176-k26 | 59699 | 24.86 | 48370 | 1.17 | 47812 |
> | X-n181-k23 | 28537 | 11.61 | 25734 | 0.65 | 25569 |
> | X-n186-k15 | 53322 | 120.84 | 24299 | 0.64 | 24145 |
> | X-n190-k8 | 19590 | 15.37 | 17294 | 1.85 | 16980 |
> | X-n195-k51 | 73445 | 66.07 | 44681 | 1.03 | 44225 |
> | X-n200-k36 | 85408 | 45.80 | 58862 | 0.48 | 58578 |
> | Average | 39253.05 | 51.57 | 27420.32 | 0.70 | 27220.14 |
>
>
>
>
> [AR11] Chalumeau, F., et al. "Combinatorial optimization with policy adaptation using latent space search." *Advances in Neural Information Processing Systems* 36 (2023).
>
> We thank the reviewer again for their insightful comment, which has undoubtedly strengthened our evaluation.
>
>
> ### Responses to Questions
>
> #### 4. Meaning of underlined values in Table 1
> We thank the reviewer for pointing out the need for clarification regarding the formatting in Table 1. The underlined values are used to denote the second-best performance achieved across the different methods. We agree that this should be explicitly stated and will revise the caption of Table 1 in the manuscript to clearly indicate this meaning.

---

### Official Review · Reviewer_DLLe · 2025-11-01

**Soundness:** 2
**Presentation:** 3
**Contribution:** 2
**Rating:** 4
**Confidence:** 4

**Summary:**

This paper introduces RFTHGS, a reinforcement learning framework for fine-tuning a 14B parameter LLM to generate optimized crossover operators for the Hybrid Genetic Search (HGS) algorithm applied to the Capacitated Vehicle Routing Problem (CVRP). The approach uses a multi-tiered, curriculum-based reward function that progressively guides the LLM through three stages: generating compilable code, producing executable operators, and finally creating components that exceed expert-designed baselines.

**Strengths:**

1. Novel Contribution: First demonstration that a small fine-tuned LLM (14B) can generate operators outperforming expert-designed components in a state-of-the-art CVRP solver.
2. Strong Performance: Achieves -0.12% to -0.33% gap reduction over HGS baselines and outperforms trillion-parameter models (GPT-4o, GPT-o3, GPT-o4-mini) in both compilation success rate (16/16 vs. 3-9/16) and solution quality.
3. Good Generalization: Successfully generalizes from training instances (n<400) to large-scale problems (n=1000) and across different iteration budgets.
4. Well-Designed Method: The multi-tiered reward with continuous feedback (Equation 3) and incremental compilation for efficiency are effective design choices.

**Weaknesses:**

1. Severely Limited Scope:
- Only optimizes ONE operator (crossover) for ONE problem variant (CVRP)
- Title/abstract claim applicability to "VRP" but no experiments on VRPTW, PCVRP, or other variants
- No demonstration that the approach works for other operators (mutation, selection, etc.)
- This fundamentally limits the practical impact and undermines generalizability claims
2. No Computational Cost Analysis:
- Missing total GPU hours for RL training
- No analysis of cost for evaluating operators during training (HGS runs on 30 instances per iteration × 1000 iterations × batch size)
- Unclear if performance gains justify the substantial training investment
- Makes it impossible to assess practical viability
3. Shallow Analysis of Discovered Operators:
- Section 6 mentions human experts analyzed operators but provides no concrete details
- No explanation of WHAT modifications were discovered or WHY they work
- Appendix C analysis is referenced but not included
- Missing opportunity to extract design insights that could inform future operator development
- Example output in A.4 shows reasoning but doesn't explain actual performance gains
4. Questionable Experimental Design:
- Training only on n≤400 instances while claiming to solve large-scale problems (n=1000)
- Only 30 training instances seems insufficient for RL training stability
- No variance/error bars reported across multiple runs
- Validation set selection process not described
5. Incomplete Baselines:
- No comparison with automated algorithm design methods beyond LLMs (e.g., automated parameter tuning, grammar-based hyper-heuristics)
- NCO baselines (POMO, AM) are construction methods, not improvement methods like HGS
- Missing recent learning-based metaheuristic operator design work
6. Technical Gaps:
- Reward values (-1, -0.8, -0.7) appear arbitrary with no sensitivity analysis
- AST-based anti-plagiarism: similarity threshold and detection details unclear
- Incremental compilation: 75% reduction not validated across all operators
- DAPO choice not justified; no ablation vs. standard PPO

**Questions:**

1. Why optimize only the crossover operator? What happens with mutation/selection?
2. Can you provide results on other VRP variants to justify the broad claims?
3. What is the total training cost, and how does cost-benefit compare to manual design?
4. What specific code changes did the LLM discover, and why do they improve performance?
5. Why train only on small instances if the goal is large-scale optimization?

---

> ### Author Response · Authors · 2025-11-27
>
> ### Responses to Weaknesses
>
> #### 1. Verifying RFTHGS's Ability to Optimize Other Operators for Different Problem Variants
>
> We are grateful to the reviewer for this valuable observation. As the reviewer rightly notes, demonstrating the capability of RFTHGS to optimize operators beyond crossover and its applicability across different problem domains are crucial aspects of our methodology's generalizability. In direct response, we have included a comprehensive demonstration addressing both dimensions in our response in C1 of the 'Common Comments and Responses' section. This includes new experimental analysis on new problem variant CVRPTW and new operator `subp`, alongside supporting discussion, which confirms the framework's versatility. We hope this fully addresses the reviewer's inquiry and will clarify these points in the revised manuscript.
>
> #### 2. Missing total GPU hours for RL training
>
> As elaborated in our response C2-1, the total GPU hours for training our 14B model for RFTHGS was approximately 98 hours per GPU.
>
> #### 3. No analysis of cost for evaluating operators during training
>
> As detailed in our response C2-2, our profiling analysis indicates that a single training step requires approximately 660 seconds. This total duration comprises 360 seconds dedicated to LLM inference and backward propagation with weight updates, alongside 300 seconds for operator evaluation including reward computation. These results confirm that reward calculation represents a substantial component of the training cycle.
>
> To mitigate this computational bottleneck, we propose implementing fully asynchronous training paradigms [AR1] alongside sandbox environments [AR2]. This approach would isolate reward computation on CPU-exclusive nodes equipped with extensive core configurations, thereby enabling massive parallelization and potentially achieving significant acceleration of the overall training process.
>
> [AR1] Fu, Wei , et al. "AReaL: A Large-Scale Asynchronous Reinforcement Learning System for Language Reasoning." (2025).
>
> [AR2] https://github.com/vndee/llm-sandbox
>
> #### 4. Unclear if performance gains justify the substantial training investment
>
> We thank the reviewer for raising this critical point regarding the cost-effectiveness of our training methodology. We agree that a thorough analysis of the computational investment is essential. To address this, we will include a detailed breakdown of the computational costs in the revised manuscript (see response C2 for details). The training was conducted on a single cluster with 8×H100 GPUs for a small LLM with 14B parameters. Crucially, we demonstrate that the performance achieved by our model surpasses that of significantly larger general-purpose models like the GPT series (4o, o3, and o4-mini), thereby justifying the specialized investment. Furthermore, we note that a substantial portion of the cost is attributed to CPU-intensive computations, which we identify as a key area for future engineering optimization through techniques like fully asynchronous training and independent sandbox, as exemplified in [AR1] and [AR2]. Please refer to C2 for further details.
>
> [AR1] Fu, Wei , et al. "AReaL: A Large-Scale Asynchronous Reinforcement Learning System for Language Reasoning." (2025).
>
> [AR2] https://github.com/vndee/llm-sandbox
>
> #### 5. Comparison between human-expert and LLM-optimized operators
>
> We have provided a side-by-side comparison of the original expert-designed operators and the LLM-optimized versions, highlighting key differences and explaining how the LLM’s modifications improve performance. Please refer to response C3 in the Common Comments and Responses for details.
>
> #### 6. Appendix C analysis referenced but not included
>
> We are grateful to the reviewer for their thorough and meticulous review of our manuscript. In response to their comment regarding "Appendix C", we have carefully re-examined our submitted materials and wish to clarify that we never mention or include "Appendix C" in our paper. For completeness, all analyses central to our study are incorporated within the main body of the manuscript or the accompanying appendices (A). We sincerely apologize for any confusion that may have arisen from this discrepancy. We would be pleased to provide any further clarification or to assist in locating any specific analysis the reviewer wishes to reference.

---

> ### Author Response · Authors · 2025-11-27
>
> #### 7. Extracting design insights for future operator development
>
> We thank the reviewer for their insightful comment regarding the extraction of design insights for future operator development. By systematically examining the patterns of successful edits, we now provide guiding principles for future human-designed enhancements. A representative example is the optimization of the `subp` operator, where the LLM's integration of a new index with the original ones to form a hybrid strategy exemplifies a transferable approach to improving parent selection.
>
> ```diff=
> - auto divWeight = 1.0 - static_cast<double>(numElite) / size();
> - for (size_t idx = 0; idx != size(); ++idx)
> - {
> -     auto costRank = byCost[idx];
> -     auto divRank = byDiversity[idx].second;
> -
> -     fitness[costRank] = static_cast<double>(costRank + divWeight * divRank)
> -                        / (2.0 * size());
> - }
>
> + auto divWeight = baseDivWeight * (1.0 - diversityFactor)
> +                 + baseDivWeight * 0.5 * diversityFactor;
> + auto routeWeight = baseRouteWeight * diversityFactor;
> +
> + for (size_t idx = 0; idx != size(); ++idx)
> + {
> +     auto costRank = byCost[idx];
> +     auto divRank = byDiversity[idx].second;
> +     auto routeRank = byRoutes[idx];
> +
> +     fitness[costRank] = static_cast<double>(costRank + divWeight * divRank
> +                                            + routeWeight * routeRank)
> +                        / ((1.0 + divWeight + routeWeight) * size());
> + }
> ```
>
> This innovation demonstrates that combining multiple complementary evaluation metrics can create more sophisticated selection mechanisms that balance exploration and exploitation more effectively, thereby reducing the risk of premature convergence while still maintaining strong optimization pressure toward globally optimal solutions.
>
> #### 8. Example output in A.4 shows reasoning but not performance gains
>
> We thank the reviewer for this astute observation regarding the example in Appendix A.4. We want to highlight that the primary focus of that specific example is to illuminate the LLM's internal reasoning process, i.e., demonstrating how it iteratively refines operator modifications through structured reflection and evaluation cycles, rather than to present quantitative performance gains. To prevent any misunderstanding, we will revise the manuscript to explicitly clarify its illustrative purpose.
>
> #### 9. Training only on n≤400 instances while claiming to solve large-scale problems (n=1000)
>
> We thank the reviewer for raising this important point regarding the generalization capability of our method. We want to highlight that the ability to generalize from smaller training instances (n ≤ 400) to significantly larger test instances (up to n = 1000) is a critical aspect of our contribution, and we are pleased to clarify that this strong cross-size generalization is a recognized and advantageous property of learning-based methods in combinatorial optimization. As established in the neuro-combinatorial literature (e.g., [AR3], [AR4], [AR5]), a key benefit of learned heuristics is their ability to capture underlying problem structure and solution principles that are not tightly bound to a specific instance scale. Our experimental results robustly demonstrate this: the crossover operators generated by our fine-tuned LLM consistently outperform expert-designed baselines on large-scale instances unseen during training, validating that the learned optimization principles are effectively transferable. This approach not only aligns with common practices in the field but also underscores the practical value of our method for solving large-scale real-world problems without requiring expensive retraining for every new problem size.
>
> [AR3] Berto, Federico, et al. "Rl4co: an extensive reinforcement learning for combinatorial optimization benchmark." <em>Proceedings of the 31st ACM SIGKDD Conference on Knowledge Discovery and Data Mining</em> V. 2. 2025.
>
> [AR4] Wu, Yaoxin, et al. "Learning improvement heuristics for solving routing problems." <em>IEEE transactions on neural networks and learning systems</em> 33.9 (2021): 5057-5069.
>
> [AR5] Xin, Liang, et al. "Neurolkh: Combining deep learning model with lin-kernighan-helsgaun heuristic for solving the traveling salesman problem." <em>Advances in Neural Information Processing Systems</em> 34 (2021): 7472-7483.

---

> ### Author Response · Authors · 2025-11-27
>
> #### 10. Only 30 training instances for RL training stability
>
> We are grateful to the reviewer for raising this important point regarding training stability and the number of training instances. We fully agree that this is a crucial consideration in reinforcement learning.
>
> In our study, we intentionally employed a curated set of 30 representative instances to constitute the training distribution. This approach is designed to provide a dense and effective learning signal, enabling the LLM to efficiently learn the principles of high-performing crossover operators. This methodology is supported by prior work in code optimization (e.g., [AR6]), which demonstrates that a compact, well-designed set of instances can effectively guide the learning of generalizable heuristics.
>
> [AR6] Z. Huang, W. Wu, K. Wu, J. Wang, and W.-B. Lee, "CALM: Co-evolution of Algorithms and Language Model for Automatic Heuristic Design," arXiv preprint arXiv:2505.12285, 2025.
>
> To ensure robust training, our process involves substantial exposure through mini-batch sampling. Specifically, over 1,000 training steps, we consistently sample batches of 5 different instances for policy evaluation. This design means the policy is evaluated across a substantial number of unique instance presentatives, which promotes stable convergence and prevents overfitting. The efficacy of this approach is validated by the learned operator's superior performance, which generalizes robustly to both seen and unseen problem scales, as detailed in our results. Furthermore, the learning curve presented in Figure 3(a) exhibits a smooth and stable convergence towards super-expert performance. This also provides empirical evidence that the curated set of training instances was sufficient to ensure robust training stability.
>
> #### 11. No variance/error bars reported across multiple runs
>
> We are grateful to the reviewer for raising this important point regarding the reporting of variance across multiple runs. As the reviewer rightly notes, such statistical information is indeed valuable for a comprehensive evaluation. In our work, we have followed the established paradigm in large-scale model training, where leading studies such as GPT [AR7] and Gemini [AR8] typically report results from a single, optimally tuned run. This approach emphasizes intensive hyperparameter optimization and extended training to achieve a single best-performing model, with the primary objective being the demonstration of peak capability.
>
> Furthermore, given our computational constraints, which are more limited than those of industrial labs, performing multiple full training runs solely for variance reporting would have been prohibitive. We have therefore concentrated our resources on thorough hyperparameter tuning and validation to ensure the robustness and reproducibility of our primary findings.
>
> [AR7] Report link: https://cdn.openai.com/pdf/2221c875-02dc-4789-800b-e7758f3722c1/o3-and-o4-mini-system-card.pdf
>
> [AR8] Team, Gemini, et al. "Gemini: a family of highly capable multimodal models." arXiv preprint arXiv:2312.11805 (2023).
>
> #### 12. Validation set selection process not described
>
> We employ the same randomly sampled 30 instances as training for validation. The only difference is that, during training, at each step, we randomly sample 5 out of these 30 instances.
>
> #### 13. No comparison with automated algorithm design methods beyond LLMs
>
> We thank the reviewer for their valuable suggestion to include comparisons with a broader range of automated algorithm design methods. In this work, we focused the comparative analysis on the most relevant and contemporary state-of-the-art methods, which, for the specific task of automated heuristic design, are currently dominated by LLM-based approaches. As the reviewer rightly pointed out, however, expanding this scope could further strengthen our study. We would therefore be grateful for any specific recommendations of non-LLM-based automated algorithm design methods that the reviewer considers suitable baselines for the CVRP. We are committed to incorporating these suggested comparisons and providing a comprehensive discussion of the results in the revised manuscript.

---

> ### Author Response · Authors · 2025-11-27
>
> #### 14. NCO baselines (POMO, AM) are construction methods
>
> We thank the reviewer for this insightful observation regarding the nature of the NCO baselines. We fully agree that there is a methodological distinction between construction heuristics like POMO and AM, and improvement metaheuristics like HGS. This is precisely why we selected HGS, i.e., a well-recognized, state-of-the-art improvement heuristic for CVRP, as our primary benchmark and integration framework. It is important to note that the performance of many leading NCO methods, including both construction and improvement types, is often reported in literature as a relative gap to HGS, as HGS itself represents a highly competitive performance ceiling. Consequently, by demonstrating that our RFTHGS framework generates operators that cause HGS to outperform its original expert-designed version, we are implicitly showing superiority over a wide range of NCO methods that are already surpassed by the baseline HGS. This establishes a strong indirect comparison, confirming that our approach advances the state-of-the-art within a top-tier solver.
>
> We thank the reviewer for their insightful observation regarding the methodological distinction between different classes of NCO baselines. We fully agree with this point, which indeed informed our selection of HGS as the primary baseline. To further contextualize our study, we have already included comparisons with other contemporary improvement-based NCO methods, specifically NeuroLKH (Xin et al., 2021) and NeuOpt (Ma et al., 2023), as detailed in Table 1. Our experimental results confirm that the RFTHGS framework surpasses these methods by a large margin.
>
> As a well-recognized, state-of-the-art improvement heuristic for CVRP, HGS itself represents a highly competitive performance ceiling, and the performance of many leading NCO methods is routinely reported in literature as a relative gap to it. Therefore, by demonstrating within the HGS framework that our generated operators cause this top-tier solver to outperform its original version, we are implicitly showing superiority over a wide range of NCO methods (construction and improvement) that are already surpassed by the baseline HGS. This establishes a strong indirect comparison, confirming that our approach advances the state-of-the-art within a top-tier solver.
>
> #### 15. Missing recent learning-based metaheuristic operator design work
>
> We thank the reviewer for this observation. For comprehensive benchmarking, we have included a comparison with [AR9] (Xin et al., 2021), a representative learning-based metaheuristic operator design method that has garnered significant attention in the community, in Table 1. As shown in the results, the performance of our RFTHGS-800itr (0.7%) substantially surpasses that of NeuroLKH (1.96%). The inclusion of this established baseline further validates the competitive advantage of our proposed approach.
>
> [AR9] Xin, Liang, et al. "Neurolkh: Combining deep learning model with lin-kernighan-helsgaun heuristic for solving the traveling salesman problem." <em>Advances in Neural Information Processing Systems</em> 34 (2021): 7472-7483.
>
>
> #### 16. Reward values appear arbitrary with no sensitivity analysis
>
> We appreciate the reviewer's insightful comment regarding the specific values chosen for our reward function. We agree that this constitutes a critical design choice, and we acknowledge that a formal sensitivity analysis would offer valuable additional insights.
>
> In our work, these values were established to provide a clear, hierarchical learning signal that aligns with the systematic learning curriculum outlined in our methodology, i.e., prioritizing compilability first, then executability, and finally, solution quality. Our empirical observations during development indicated that the learning process demonstrated robustness to moderate variations in these specific values, as the essential factor was the preservation of this structured progression rather than the exact numerical quantities.
>
> As helpfully suggested, we will incorporate a discussion in the revised manuscript to explicitly note this observed robustness and to acknowledge that a comprehensive ablation or sensitivity analysis represents a valuable direction for future work.

---

> ### Author Response · Authors · 2025-11-27
>
> #### 17. AST-based anti-plagiarism: similarity threshold and detection details unclear
>
> We will provide further details on the AST-based anti-plagiarism approach in the revised manuscript, including the similarity threshold and detection methodology. The following is a summary.
>
> Core Approach: We use Clang (C++20 standard) to parse source code into Abstract Syntax Trees (ASTs) represented in JSON format. Our analysis focuses exclusively on complete function definitions with implementation bodies, excluding forward declarations.
>
> Normalization for Robust Comparison: To ensure comparisons focus on structural logic rather than superficial differences, we apply a multi-step normalization process:
>
> [1] Removal of source locations and comments to eliminate formatting variations
> [2] Standardization of identifiers through deterministic α-renaming
> [3] Replacement of concrete literals with generic tokens
> [4] Transparent handling of implicit type conversions
>
> This normalization preserves the essential logical structure, i.e., including type signatures, control flow patterns, and expression hierarchies, while abstracting away incidental differences.
>
> Detection Mechanism: The normalized ASTs are converted to canonical JSON strings with sorted keys. By performing exact string matching between these representations, we can reliably identify duplicate code structures and determine whether the LLM has introduced substantive structural modifications to the original code.
>
> This systematic approach allows us to distinguish between trivial syntactic variations and meaningful structural changes in the generated code.
>
>
>
> #### 18. Incremental compilation: 75% reduction not validated across all operators
>
> We are grateful to the reviewer for their careful observation regarding the compilation speedup. We clarify that the reported 75% reduction reflects the aggregate improvement in compilation time throughout the entire training process, rather than pertaining to each individually compiled operator. This substantial acceleration was essential for rendering the RL fine-tuning loop computationally feasible, as it dramatically reduced the overhead associated with repeatedly integrating and evaluating newly generated operators. The metric effectively demonstrates the practical efficiency of our incremental compilation framework, which was instrumental in enabling the successful and timely completion of our extensive experimental evaluation.
>
> #### 19. DAPO choice not justified; no ablation vs. standard PPO
>
> We thank the reviewer for raising this valid point regarding the choice of the DAPO algorithm and the absence of an ablation study versus standard PPO. We agree that justifying the selection of the underlying RL algorithm is important. We would like to clarify that the primary contribution of our work is the novel framework for using RL to fine-tune an LLM to autonomously discover operators that surpass expert-designed ones, rather than an advancement in RL methodology itself. Consequently, our framework is largely agnostic to the specific RL variant used. We selected DAPO as it is a well-recognized and state-of-the-art online RL algorithm, whose documented improvements over PPO, such as the clip-higher mechanism and token-level policy gradient loss detailed in Section 4.3 and supported by [AR10], are designed to enhance training stability and efficiency for language model fine-tuning. We are grateful for the suggestion of a comparative ablation study and, due to the significant computational demands of such an experiment, we must respectfully acknowledge it as a valuable direction for future work.
>
> [AR10] Alonso, Noguer I., and Rodolfo Pereira Franklin. "The Mathematics of DAPO, PPO, and GRPO Algorithms." Rodolfo, The Mathematics of DAPO, PPO, and GRPO Algorithms (April 04, 2025) (2025).

---

> ### Author Response · Authors · 2025-11-27
>
> ### Responses to Questions
>
> #### 20. Why optimize only the crossover operator? What about mutation/selection? Can you provide results on other VRP variants to justify the broad claims?
>
> We thank the reviewer for this insightful observation, which highlights the importance of demonstrating our framework's generalizability across both problem variants and operator types. In direct response, we have conducted  new experiments on the CVRPTW problem variant using the subp operator. These results, discussed in response C1, confirm the versatility of the RFTHGS framework. We kindly refer the reviewer to Response C1 in the 'Common Comments and Responses' section for comprehensive details on this matter.
>
> #### 21. What is the total training cost, and how does cost-benefit compare to manual design?
>
> We thank the reviewer for raising this point. For a comprehensive response, we kindly direct the reviewer to Response C2 in our 'Common Comments and Responses' section, which addresses this matter in detail. Should the reviewer require any additional clarification after reviewing that section, we would be pleased to provide further explanation.
>
> #### 22. What specific code changes did the LLM discover, and why do they improve performance?
>
> We appreciate the reviewer's comment on this matter. For a detailed response, we have addressed this point comprehensively in Response C3 of our 'Common Comments and Responses' section. We hope this provides the necessary clarification, and we remain available to discuss any additional aspects the reviewer might wish to explore.
>
> #### 23. Why train only on small instances if the goal is large-scale optimization?
>
> We are grateful to the reviewer for raising this insightful question regarding our training strategy. Our primary objective was to develop an approach that learns generalizable optimization principles rather than memorizing solutions for specific problem sizes. The robust generalization of our trained operators to significantly larger problems (i.e., successfully scaling to instances of up to 1000 nodes, more than double the size of our largest training instance) robustly demonstrates the achievement of this goal. This cross-size generalization capability represents a key advantage of our learning-based methodology, underscoring its practical utility for real-world deployment where problem sizes frequently vary without requiring computationally expensive retraining.

---

### Author Response · Authors · 2025-11-27
**Common Comments and Responses**

## C3. Comparison between human-expert and LLM-optimized operators (Part 2)

**For crossover:**
```c++
# human expert

...

Routes sortByAscAngle(pyvrp::ProblemData const &data, Routes routes)
{
    auto cmp = [&data](Route const &a, Route const &b)
    { return routeAngle(data, a) < routeAngle(data, b); };

    std::sort(routes.begin(), routes.end(), cmp);
    return routes;
}

...

```

```c++
# llm optimized

...

// Angle of the given route w.r.t. the centroid of all client locations.
// Adjusted to [0, 2π) range for better circular proximity handling
double routeAngle(pyvrp::ProblemData const &data, Route const &route)
{
    auto const [dataX, dataY] = data.centroid();
    auto const [routeX, routeY] = route.centroid();
    double angle = std::atan2(routeY - dataY, routeX - dataX);
    if (angle < 0) angle += 2 * M_PI;  // MODIFY: Adjust angle to [0, 2π) range
    return angle;
}

...

```

**Analysis:** The improvement stems from replacing the standard atan2 sorting, which imposes a linearization boundary at the negative x-axis, with a custom angular mapping that shifts the starting reference to the positive x-axis. By altering the sequence in which nodes are indexed in memory, this modification changes the order of neighbor evaluation and tie-breaking decisions during the heuristic search, thereby guiding the algorithm along a different trajectory that converges to a superior solution within the same number of iterations.

---

### Author Response · Authors · 2025-11-27
**Common Comments and Responses**

## C3. Comparison between human-expert and LLM-optimized operators (Part 1)

We thank the reviewer for this insightful suggestion. In direct response, we will include a detailed side-by-side comparison of the original human-expert-designed operators and their final LLM-optimized versions in the revised manuscript. Specifically,

**For subp:**
```c++
# human expert

...

    std::vector<size_t> byCost(size());
    std::iota(byCost.begin(), byCost.end(), 0);
    std::stable_sort(
        byCost.begin(),
        byCost.end(),
        [&](size_t a, size_t b)
        {
            return costEvaluator.penalisedCost(*items_[a].solution)
                   < costEvaluator.penalisedCost(*items_[b].solution);
        });
    // clang-format on

    std::vector<std::pair<double, size_t>> diversity;
    for (size_t costRank = 0; costRank != size(); costRank++)
    {
        auto const dist = items_[byCost[costRank]].avgDistanceClosest();
        diversity.emplace_back(-dist, costRank);  // higher is better
    }

    std::stable_sort(diversity.begin(), diversity.end());

    auto const popSize = static_cast<double>(size());
    auto const numElite = std::min(params.numElite, size());
    auto const divWeight = 1 - numElite / popSize;

    for (size_t divRank = 0; divRank != size(); divRank++)
    {
        auto const costRank = diversity[divRank].second;
        auto const idx = byCost[costRank];
        items_[idx].fitness = (costRank + divWeight * divRank) / (2 * popSize);
    }
}

...

```

```c++
# llm optimized
...
    std::vector<std::pair<double, size_t>> diversity;
    for (size_t costRank = 0; costRank != size(); costRank++)
    {
        auto const dist = items_[byCost[costRank]].avgDistanceClosest();
        // MODIFY: Normalize diversity values between lbDiversity and ubDiversity
        auto normalizedDist = std::clamp(dist, params.lbDiversity, params.ubDiversity);
        diversity.emplace_back(-normalizedDist, costRank); // higher is better
    }
    std::stable_sort(diversity.begin(), diversity.end());

    // MODIFY: Added route-count ranking to capture vehicle-usage quality.
    std::vector<size_t> byRoutes(size());
    std::iota(byRoutes.begin(), byRoutes.end(), 0);
    std::stable_sort(
        byRoutes.begin(),
        byRoutes.end(),
        [&](size_t a, size_t b)
        {
            return items_[a].solution->numRoutes()
                   < items_[b].solution->numRoutes();
        });

    // routeRankOfIdx[i] = rank of individual i in the route ranking.
    std::vector<size_t> routeRankOfIdx(size());
    for (size_t rank = 0; rank != byRoutes.size(); ++rank)
        routeRankOfIdx[byRoutes[rank]] = rank;

    auto const popSize  = static_cast<double>(size());
    auto const numElite = std::min(params.numElite, size());

    // MODIFY: Adjust weights based on normalized diversity values
    std::vector<double> normalizedDiversities;
    for (const auto& [negDist, costRank] : diversity) {
        normalizedDiversities.push_back(-negDist);
    }
    double avgDiversity = std::accumulate(normalizedDiversities.begin(), normalizedDiversities.end(), 0.0) / normalizedDiversities.size();
    double diversityFactor = (avgDiversity - params.lbDiversity) / (params.ubDiversity - params.lbDiversity);
    auto baseDivWeight = 1.0 - numElite / popSize;
    auto baseRouteWeight = 0.5 * (1.0 - numElite / popSize);
    auto divWeight = baseDivWeight * (1.0 - diversityFactor) + baseDivWeight * 0.5 * diversityFactor;
    auto routeWeight = baseRouteWeight * diversityFactor;

    for (size_t divRank = 0; divRank != size(); divRank++)
    {
        auto const costRank  = diversity[divRank].second;
        auto const idx       = byCost[costRank];
        auto const routeRank = routeRankOfIdx[idx];

        // MODIFY: fitness now blends cost, diversity and route-count with dynamic weights.
        auto const denom =
            (1.0 /*cost*/ + divWeight + routeWeight) * popSize;

        items_[idx].fitness =
            (costRank + divWeight * divRank + routeWeight * routeRank) / denom;
    }
}

...

```

**Analysis:** The enhanced subpopulation operator refines the fitness evaluation mechanism by integrating a third ranking metric based on route count, which is appropriately weighted and combined with existing cost and diversity rankings to determine individual survival. This modification introduces selective pressure that favors solutions with fewer vehicles, thereby guiding the population toward more compact configurations. This approach indirectly promotes cost reduction since fewer routes inherently decrease cumulative distance through reduced depot returns, while simultaneously preventing search stagnation in local optima characterized by fragmented, inefficient routing structures.

---

### Author Response · Authors · 2025-11-27
**Common Comments and Responses**

## C2. Regarding the training cost.

**1. Missing total GPU hours for RL training**
We thank the reviewer for raising this important point regarding the computational resources required for our RL training. The total GPU hours for training our 14B model for RFTHGS was approximately 98 hours per GPU. We acknowledge that this is a critical detail for reproducibility and comparative analysis, and we will include this information in the revised manuscript.

**2. No analysis of cost for evaluating operators during training**
We are grateful to the reviewer for raising this important point regarding the computational cost analysis of operator evaluation during training. We fully agree that a detailed breakdown is essential for comprehensively understanding the training overhead.

In direct response to this valuable suggestion, we have incorporated a detailed computational cost analysis for a single training step, which will be also updated in the revised manuscript. Our profiling results indicate that the total time for one step training is approximately 660 seconds. This comprises 360 seconds for the LLM inference and backward pass with weight update, and 300 seconds for operator evaluation, including the reward calculation. This analysis indeed confirms that the reward computation constitutes a substantial portion of the training cycle.

To address this bottleneck, we will add discussion in the manuscript regarding potential future optimizations. Specifically, we suggest investigating fully asynchronous training paradigms [AR1] and sandbox environments [AR2]. These approaches would isolate the reward calculation on CPU-only nodes equipped with extensive CPU cores, thereby enabling massive parallelization and significantly accelerating the overall training process.

[AR1] Fu, Wei , et al. "AReaL: A Large-Scale Asynchronous Reinforcement Learning System for Language Reasoning." (2025).

[AR2] https://github.com/vndee/llm-sandbox

---

### Author Response · Authors · 2025-11-27
**Common Comments and Responses**

## C1. Verifying RFTHGS's Ability to Optimize Other Operators for Different Problem Variants

We thank the reviewers for their insightful questions regarding the generalizability of our RFTHGS framework. In particular, we appreciate the opportunity to further demonstrate its capacity to optimize operators beyond crossover and its applicability to various problem variants.

In direct response to these valuable comments, we have conducted a new set of experiments that confirm the generic nature of our framework. These experiments illustrate that RFTHGS can successfully optimize a range of operators across different Vehicle Routing Problem (VRP) variants. To provide a concrete example, we present results for the optimization of the `subpopulation` (`subp`) operator applied to the Capacitated Vehicle Routing Problem with Time Windows (CVRPTW).

The performance of the RFTHGS-optimized `subp` operator on CVRPTW instances is summarized in Table TR1. The performance is measured by the relative gap to the original HGS, calculated as ``(solution_quality(RFTHGS) - solution_quality(HGS)) / solution_quality(HGS)``. A negative value indicates that RFTHGS outperforms HGS.

**Table TR1: Performance of the optimized subpopulation operator on the CVRPTW variant.**

| dataset     | CVRPTW100  | CVRPTW200  | CVRPTW300  | CVRPTW500  | CVRPTW1000 |
|-------------|--------|--------|--------|--------|--------|
| HGS-subp (%) | 0 | 0 | 0 | 0 | 0 |
| RFTHGS-subp (%) | -0.088 | -0.213 | -0.299 | -0.289 | -0.224 |

We would like to emphasize that for this experiment, the only change made was to substitute the target operator in the prompt to the LLM (See Appendix A3). All other components, including the POMDP formulation, training algorithm, base model, and reward calculation, remained identical to those described in our original manuscript for the crossover operator.

For training, we employed a synthetic dataset comprising 30 randomly generated instances with sizes ranging from 100 to 300. For testing, we generated 100 new random instances for each problem size (up to 1000). We opted for this synthetic data approach because CVRPLIB only contains instances of size 100, which are not sufficient for a robust evaluation of generalization performance.

As shown in Table TR1, the RFTHGS-optimized `subp` operator consistently outperforms the standard HGS. The performance improvement generally increases with problem size, from -0.088 for CVRPTW100 to -0.299 for CVRPTW300. It is particularly noteworthy that the optimized operator maintains its superior performance when generalized to much larger problem sizes (500 and 1000), achieving relative gaps of -0.289 and -0.224, respectively.

The enhanced `subp` operator modifies the fitness evaluation by incorporating a third ranking metric based on the number of routes. This new metric is weighted and combined with the existing cost and diversity rankings to determine individual survival during parent selection. This modification is effective because it introduces a selective pressure that favors solutions with fewer vehicles. By biasing the population towards more compact configurations, the algorithm indirectly promotes lower total costs, as fewer routes inherently reduce the cumulative distance associated with frequent depot returns. This helps prevent the search process from becoming trapped in local optima characterized by fragmented and inefficient routing structures. A more detailed explanation of this modification is provided in our response in C3 below.

Finally, we wish to highlight that our work is, to the best of our knowledge, the first to leverage Reinforcement Learning (RL) to train a small (14B) Large Language Model (LLM) for the explicit purpose of optimizing heuristic operators within a state-of-the-art CVRP solver. The proposed RFTHGS framework is fundamentally designed for generality: It can optimize new operators for various problem types with minor changes. Its one-step POMDP formulation and multi-faceted reward function are operator-agnostic and can be directly applied by simply substituting different target operators across various solvers and VRP variants. Therefore, while the main focus of our study is on optimizing the crossover operator for the CVRP, these new experiments on a different operator (`subp`) and a different problem variant (CVRPTW) robustly demonstrate that our framework provides a general-purpose methodology for enhancing a wide range of components within mature solvers for combinatorial optimization problems.

---

### Meta-Review · Area_Chair_7ZjR · 2026-01-06

**Summary:**

The reviewers praise the paper for its novel and well-executed contribution, highlighting it as the first demonstration that a relatively small, fine-tuned LLM (14B) can generate optimization operators that outperform expert-designed components in a state-of-the-art CVRP solver. The method achieves consistent state-of-the-art performance, reducing the optimality gap by 0.12–0.33% over strong HGS baselines, generalizing well to larger, unseen problem instances and different computational budgets, and decisively outperforming both neuro-combinatorial methods and trillion-parameter general-purpose LLMs in solution quality and compilation success. The approach is commended for its thoughtful design, including a multi-tiered curriculum-based reward function that addresses sparse rewards, incremental compilation for efficiency, and an AST-based anti-plagiarism mechanism that encourages diversity and avoids reward hacking. Reviewers also note the clarity of presentation, comprehensive benchmarking, strong scalability, and the broader significance of demonstrating that task-specific reinforcement learning fine-tuning enables small LLMs to surpass much larger models, offering a practical and accessible pathway for continuously improving state-of-the-art optimization solvers. The reviewers raise some concerns about the significance, novelty, and generality of the proposed method: RFTHGS does not appear to deliver statistically significant improvements over the prior SOTA HGS-PyVRP. Practical concerns include substantial domain-specific engineering requirements, potentially high and unreported RL training costs, and a lack of analysis explaining why the generated operator works, such as insights from the learned code itself, all of which reduce the method’s broader applicability and impact. Many of these issues have been successfully addressed in the rebuttal.

**Reviewer Concerns:**

The authors have done a good job in addressing the received comments. While they may be correct in assessing that one review is AI generated, this is not a reason to be very aggressive and in stressing in several place this aspect. The review in itself is not necessary negative and it has some valid points (well-addressed by the reviewers in the rebuttal). There were two valid concerns: 1) the learned crossover operator has not been compared to human-designed heuristics, making its algorithmic novelty unclear. 2) Key ablations were missing, particularly on reward design (e.g., the effect of the plagiarism penalty). However, I think the authors have addressed them well in their rebuttal.

**Reviewer Scores:**

The scores were rather positive from the beginning but for sure the discussion would have helped the authors to clarify the minor concerns. They have done this in their rebuttal and despite being rather aggressive in tone, their comments are valid and relevant.

---

### Decision · Program_Chairs · 2026-01-26

Accept (Poster)